# Iron-coated Komodo dragon teeth and the complex dental enamel of carnivorous reptiles

Aaron R. H. LeBlanc [1] ✉, Alexander P. Morrell [1], Slobodan Sirovica[1,2], Maisoon Al-Jawad[3], David Labonte[4], Domenic C. D'Amore[5], Christofer Clemente [6], Siyang Wang [7], Finn Giuliani[7], Catriona M. McGilvery[7], Michael Pittman [8], Thomas G. Kaye [9], Colin Stevenson[10], Joe Capon[11], Benjamin Tapley[11], Simon Spiro[12] & Owen Addison[1,13]

Komodo dragons (*Varanus komodoensis*) are the largest extant predatory lizards and their ziphodont (serrated, curved and blade-shaped) teeth make them valuable analogues for studying tooth structure, function and comparing with extinct ziphodont taxa, such as theropod dinosaurs. Like other ziphodont reptiles, *V. komodoensis* teeth possess only a thin coating of enamel that is nevertheless able to cope with the demands of their puncture–pull feeding. Using advanced chemical and structural imaging, we reveal that *V. komodoensis* teeth possess a unique adaptation for maintaining their cutting edges: orange, iron-enriched coatings on their tooth serrations and tips. Comparisons with other extant varanids and crocodylians revealed that iron sequestration is probably widespread in reptile enamels but it is most striking in *V. komodoensis* and closely related ziphodont species, suggesting a crucial role in supporting serrated teeth. Unfortunately, fossilization confounds our ability to consistently detect similar iron coatings in fossil teeth, including those of ziphodont dinosaurs. However, unlike *V. komodoensis*, some theropods possessed specialized enamel along their tooth serrations, resembling the wavy enamel found in herbivorous hadrosaurid dinosaurs. These discoveries illustrate unexpected and disparate specializations for maintaining ziphodont teeth in predatory reptiles.

Ziphodont teeth evolved convergently in several extinct apex predators, including the 'pelycosaur' *Dimetrodon*[1] and theropod dinosaurs[1–6]. These teeth have serrated cutting edges composed of dentine-cored serrations (denticles) capped with a thin veneer of hard enamel. The slenderness of the hard enamel cap is most striking in extinct ziphodont species, where even the largest predatory dinosaurs have only 10–20% of the absolute enamel thickness of a human tooth and lack the structural complexity of mammalian enamel[2,7]. Despite having thin, structurally simpler enamel, amniotes repeatedly evolved ziphodont teeth[2], suggesting that this tooth morphology is linked to the success of many extinct carnivores. This also suggests that there may be common specializations hidden in ziphodont enamel which enable these teeth to effectively cut and tear into animal tissue.

Unfortunately, our poor understanding of the material properties of non-mammalian dental tissues in general[7,8] and of the

**Fig. 1 | Pigmented cutting edges in *V. komodoensis* teeth. a**, Lateral view of the skull of *V. komodoensis* (Natural History Museum, London, NHMUK 1934.9.2.1). **b**, Lingual view of a dentary tooth position showing several unerupted replacement teeth with orange pigmentation (American Museum of Natural History, AMNH 37912). **c**, Dorsal view of two erupted teeth from a fluid-preserved specimen (Zoological Society of London) showing pigmented cutting edges and apices. **d**, White light (WL) image of an erupted and unerupted tooth in the same specimen. **e**, Laser stimulated fluorescence (LSF) image of the same specimen, showing the pronounced serration pigmentation in both the erupted and unerupted teeth. **f**, Dorsal view of three left dentary teeth in NHMUK 1934.2.1 showing identical pigmentation on the tooth apices and mesial serrations.
**g**, Lateral view of an isolated replacement tooth (Museum of Life Sciences, MoLS X263). **h**, Close-up of tooth apex in **g** showing the orange pigmentation along the tooth tip. **i**, Distal view of tooth serrations of MoLS X263 showing orange serrations and tooth apex. **j**, Polished thick section through mesial denticles of a tooth (Queensland Museum, Australia, J94036-2) showing orange pigmentation restricted to the enamel. **k**, SEM image of three mesial serrations of J94036-2. **l**, Close-up of serration enamel showing the bright coating. **m**, Close-up of the crown apex enamel showing the same nanocrystalline coating. Asterisks indicate pigmented regions. de, dentine; en, enamel; et, erupted tooth; gi, gingiva; ut, unerupted tooth.

confounding effects of fossilization, limit our ability to infer structure–function relationships in extinct lineages. However, the Komodo dragon (*Varanus komodoensis*) is a rare example of an extant reptile bearing ziphodont teeth and is, therefore, a key taxon for understanding their function[5,9–12]. Here, we describe the unique specializations within the enamel of *V. komodoensis* and compare them with those of other squamates, crocodylians and theropod dinosaurs. Through state-of-the-art elemental, structural and nanomechanical characterization, we (1) reveal surprising structural and chemical complexity of *V. komodoensis* teeth; (2) compare adaptations for maintaining cutting-edge enamel in varanids and extant crocodylians; and (3)

contrast these adaptations in extant reptiles with those of extinct theropod dinosaurs.

## Results

### Iron-coated Komodo dragon teeth

Our initial examination of the teeth of skeletal and fluid-preserved specimens of *V. komodoensis* revealed an overlooked feature of their dentition: all the unworn tooth crown tips and serrations are pigmented orange (Fig. 1 and Supplementary Table 1). Erupted and unerupted teeth are equally pigmented, indicating that it is not staining from feeding (Fig. 1 and Supplementary Fig. 1). The orange

colouration is most intense along serrations closer to the crown tips, whereas the rest of the crown is coated in transparent enamel, making these regions white from the underlying dentine. The pigmentation is even more pronounced when using laser-stimulated fluorescence (LSF)[13] to image both erupted and unerupted teeth, in which the surrounding enamel fluoresces brightly and the pigmented serrations and tips appear dark (Fig. 1d,e).

Similar enamel pigmentation is found in specialized mammal teeth[14]; however, this feature has never been reported before in a carnivorous reptile. Scanning electron microscope (SEM) imaging of acid-etched sections of the serrations and tooth apices of *V. komodoensis* showed that the enamel is only 20 µm thick (Fig. 1k–m). Moreover, it is composed of parallel-crystallite enamel; a simple, prismless enamel type where bundles of crystallites are oriented perpendicular to the outer surface (Fig. 1m). However, the outermost 1–2 µm in *V. komodoensis* enamel consists of a smooth coating which appears bright under SEM. This coating is found along the serrations and crown tips.

To determine the composition of the pigmented regions, we subjected functional and replacement teeth to several complimentary imaging and characterization techniques (Methods). The orange colouration of the enamel is similar to the iron oxide-enriched outer layers in mammals with pigmented teeth[14,15]. Indeed, every technique detected iron in the outer 1–2 µm of the enamel and occasionally an associated concentration of zinc (Supplementary Figs. 2 and 3). The high zinc content in the outermost enamel may be a product of normal enamel secretion and maturation, similar to that in mammalian enamel[16]. However, the iron was consistently concentrated along the mesial and distal serrations in horizontal sections but not along enamel further away from the serrations (Fig. 2g–i and Supplementary Fig. 2). SEM–energy-dispersive X-ray spectroscopy (EDS) and laser ablation inductively coupled plasma mass spectrometry (LA-ICP-MS) analyses confirmed that the iron-enriched layer coincides with the thin outer coating we identified along the crown tips and serrations under SEM, which overlies the calcium-rich enamel (Fig. 2j,k and Supplementary Figs. 2 and 3). The colocation of high iron and higher oxygen signals (Supplementary Fig. 2k,l) suggests an iron oxide composition which coats and colours the outer enamel orange.

We further isolated a portion of this iron coating using focused ion beam (FIB) milling to conduct higher resolution scanning transmission electron microscope (STEM) and STEM–EDS analyses to determine how this iron-rich layer is incorporated into the enamel. These data revealed that the iron-enriched layer consists of a thin (100–200 nm) inner region which is interspersed between individual enamel crystallite bundles and a thicker outer layer, devoid of calcium, coating the enamel (Fig. 2l–n).

### Iron in other extant reptile teeth

Given the surprising and consistent orange colouration of both the replacement and functional teeth in *V. komodoensis*, we further surveyed skeletal collections to determine the prevalence of this feature across a sample of *Varanus* species (Fig. 3a). Orange cutting edges were present in *V. salvadorii*, *V. rosenbergi* and *V. giganteus* to varying degrees and occasionally in *V. varius*, *V. salvator* and *V. indicus* teeth (Supplementary Fig. 4). These species all bear serrations, albeit smaller than those of *V. komodoensis*. Pigmented cutting edges were not visible in five other *Varanus* species, including some with small serrations and others that had non-serrated cutting edges. For comparisons, the cutting edges of the teeth of the anguimorph lizard *Heloderma* and a small sample of other squamate specimens we examined were not pigmented (Supplementary Fig. 5 and Supplementary Table 1). This suggests that iron-pigmented cutting edges are found in several ziphodont *Varanus* species but are most pronounced in *V. komodoensis*.

The prevalence of pigmented cutting edges, particularly in the larger serrated varanid teeth, led us to further predict that the largest ziphodont squamate, the extinct *Varanus priscus* ('Megalania') may

have also possessed iron-coated serrations. However, these fossilized teeth showed no visual evidence of pigmentation (Fig. 3b,c). Indeed, the varanid fossil record in general includes many fossilized teeth bearing serrations[17–21]; however, none of these occurrences shows any obvious pigmentation similar to their extant counterparts. Furthermore, the small absolute sizes of varanid teeth and the paucity of fossil teeth available for destructive analyses make it difficult to assess the material properties of iron coatings in extant taxa or how fossilization has hindered our ability to visually detect them in extinct species. We therefore sought to test for iron-pigmented enamel in another reptilian group with much larger teeth and a more abundant dental fossil record: modern and fossil crocodylians and the ziphodont teeth of theropod dinosaurs.

We first examined the teeth of several species of crocodylians to determine if extant toothed archosaurs are capable of sequestering iron in a similar fashion to *Varanus*. We subjected shed teeth of zoo-kept crocodylians (*Alligator mississippiensis*, *Crocodylus porosus*, *Osteolaemus tetraspis* and *Tomistoma schlegelii*) to similar elemental analyses. Although their teeth are not serrated, we observed orange-coloured carinae (non-serrated cutting edges) in some but not all crocodylian teeth we sampled. However, even in crocodylian teeth lacking obvious pigmentation under white light, we were able to identify similar fluorescence patterns along the carinae to those we identified in *V. komodoensis* teeth using LSF (Fig. 4 and Supplementary Figs. 6 and 7). LA-ICP-MS and synchrotron-based X-ray microfluorescence (S-µXRF) analyses confirmed the presence of an outer iron-enriched enamel layer in all four crocodylian species, similar in distribution to those seen in *V. komodoensis* (Fig. 4 and Supplementary Figs. 6–8). In some cases, these appeared to be more concentrated along the carinae, whereas in others the iron layer was more evenly distributed along the enamel surface. Thus, extant toothed archosaurs appear to be able to sequester iron in their enamel and to varying degrees along their cutting edges, albeit in more subtle arrangements compared with those of *V. komodoensis*.

### Composition of the iron coating in *V. komodoensis*

We conducted iron-edge X-ray absorption near-edge spectroscopy (Fe-XANES) to characterize the chemical speciation of the iron coating in *V. komodoensis* and compare it to other iron-enriched enamels. Comparison with standards suggested that the white line (7,132 eV) and post-edge features, including a peak at 7,147 eV, were most consistent with ferrihydrite (Supplementary Fig. 9), which is also found in pigmented mammal teeth[15]. Combinations with other iron oxide species cannot be excluded; however, haematite exhibits several post-edge features between 7,132 and 7,160 eV and magnetite exhibits a small shift in the principal absorption peak energy, both of which were not observed in the *V. komodoensis* iron coating. The Fe K-edge XANES spectra were also compared with an extant crocodylian tooth (*Crocodylus porosus*) and iron-enriched beaver (*Castor canadensis*) enamel. These spectra were different across taxa, suggesting variation in iron speciation across species with pigmented enamel.

### Searching for iron coatings in fossil reptile teeth

We similarly subjected Late Cretaceous-aged crocodylian teeth from Dinosaur Provincial Park (Alberta, Canada) to LA-ICP-MS and S-µXRF to examine and compare the distribution of iron and calcium between extant and fossilized crocodylian teeth. The resulting maps consistently showed increased calcium and iron signals throughout the dentine of each tooth with no indication of increased iron concentrations along the outer enamel (Fig. 4o,p and Supplementary Fig. 7). These results indicate that even if these crocodylian teeth possessed iron in their enamel, fossilization may obscure this signal, as a result of the abundance of iron and other exogenous elements in fossilized dental tissues.

We then undertook elemental imaging of several tyrannosaurid and dromaeosaurid theropod specimens (Supplementary Table 2) to

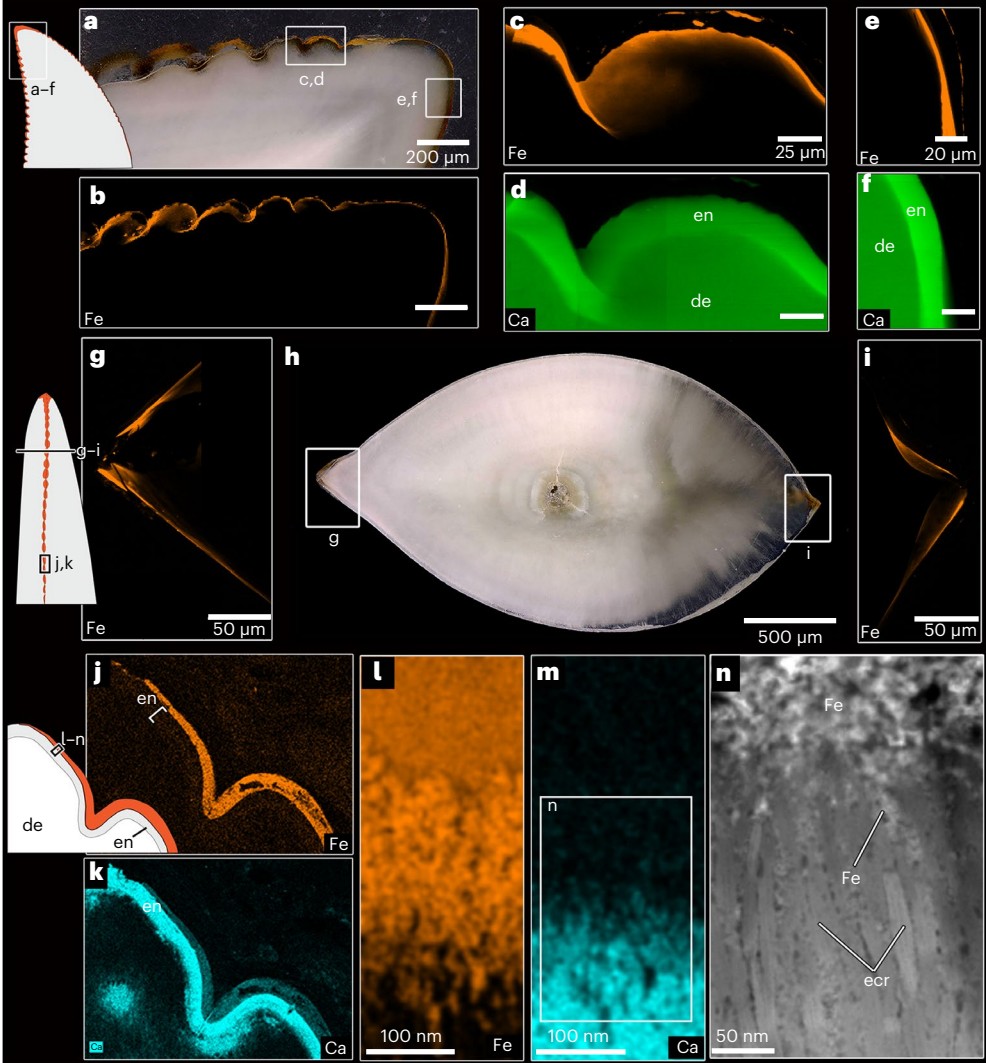

**Fig. 2 | Elemental and microstructural analyses of serration and tooth apex enamel in *V. komodoensis*. a**, Polished thick section of a tooth apex and distal serrations (MoLS X263). **b**, S-µXRF elemental map for iron along the tooth apex and serrations in MoLS X263. **c,d**, High-resolution S-µXRF map of iron along two apical serrations (**c**), compared with the calcium map of the same region (**d**), showing that iron is restricted to the outer enamel surface. **e,f**, High-resolution S-µXRF map of iron along the tip of the tooth (**e**), compared with the calcium map of the same region (**f**). **g**, High-resolution S-µXRF map of iron along a horizontal section through a distal serration in the same tooth. **h**, Wholeview image of horizontal section through MoLS X263. **i**, Higher resolution S-µXRF map of iron along the horizontal section through a mesial serration in MoLS X263. **j**, SEM-EDS map of iron from a thick section of a functional tooth (J94036-2) showing that the

iron is most abundant in the coating overlying the enamel. **k**, SEM-EDS map of calcium from the same view as in **j**, showing reduced calcium signal in the outer coating. **l**, STEM-EDS map of iron from an FIB-milled portion of the outer enamel coating of J94036-2 (outer surface of tooth is towards the top). **m**, STEM-EDS map of calcium in the same region, showing lack of calcium in the iron-rich coating. **n**, STEM image of the interface between the iron-rich coating (above) and crystalline enamel (below) and iron-rich material interspersed between enamel crystallites (middle). In all elemental maps, brighter colours indicate higher counts. In the illustrations, red represents the pigmented regions, white is dentine and grey is enamel. am, amorphous region; cr, crystalline region; ecr, enamel crystallites; edj, enamel-dentine junction; Fe, iron.

examine elemental composition and heterogeneity along their serrations. Given the ability of extant archosaurs to sequester iron in their enamel and the intensity of the colouration in ziphodont *Varanus* teeth, we expected to see some level of increased iron signal within the serration enamel of ziphodont dinosaurs. However, preliminary white light and LSF imaging of a sample of theropod teeth showed no evidence of differential fluorescence patterns along the cutting edges (Fig. 5a–f and Supplementary Fig. 10). Furthermore, LA-ICP-MS and S-µXRF analyses of theropod teeth showed similar results to those of fossil crocodylian teeth: the dentine was enriched in iron, whereas the enamel was relatively depauperate, except in regions of diagenetic crack infilling (Fig. 5 and Supplementary Figs. 10–14). A similar phenomenon occurred for other elements, which had comparable or higher signals in the dentine compared with the enamel, whereas the

opposite occurs in extant reptile teeth. These findings suggest that, similar to our crocodylian sample, either diagenetic alterations mask this feature in fossil theropod teeth or theropods did not normally sequester iron along their serrations and may have used an alternative adaptation to reinforce their serration enamel.

## Structural complexity in the enamel of theropod teeth

To assess microstructural features within theropod serrations, we next undertook qualitative analyses (light and scanning electron microscopy) to characterize theropod enamel microstructure. Compared to *V. komodoensis*, theropod enamel is often thicker (50–200 µm) and can be composed of a more complex microstructure (Fig. 6 and Supplementary Figs. 15 and 16). Tyrannosaurid enamel in particular is organized into discrete columns of herringboned stacks of crystallites, which

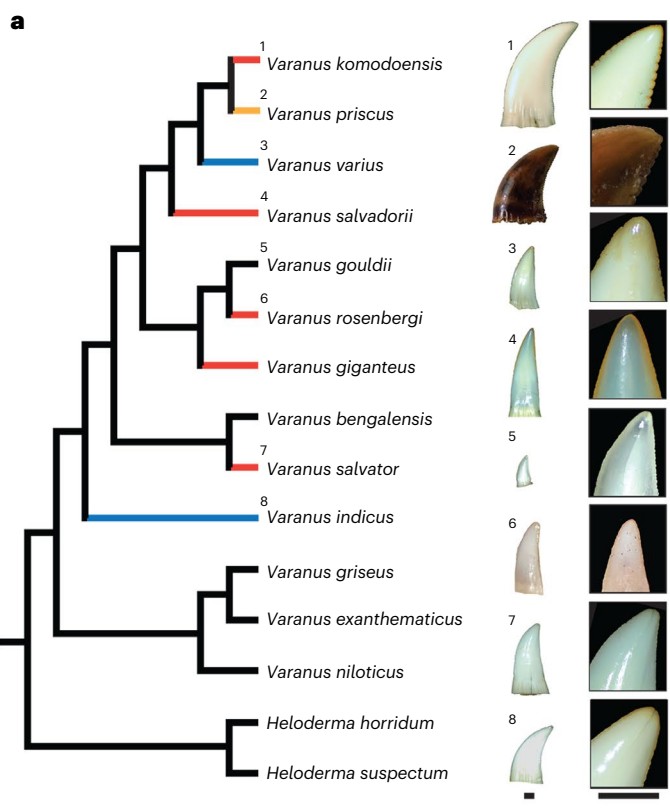

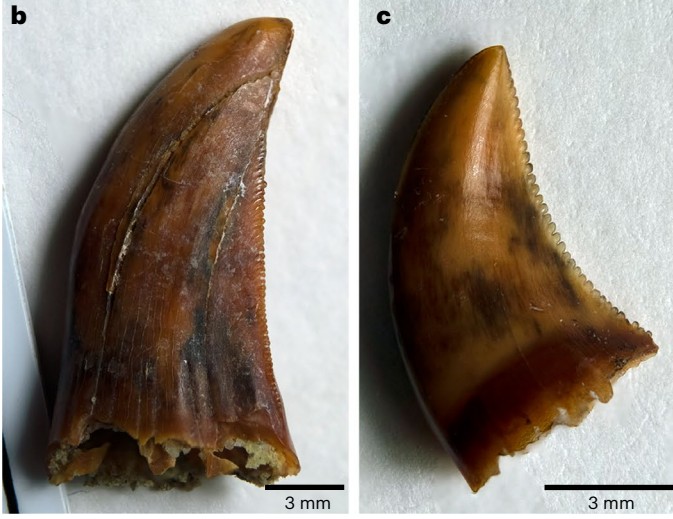

**Fig. 3 | Distribution of pigmented teeth within the genus *Varanus*.**
**a**, Simplified phylogeny of *Varanus* species examined in this study (modified from ref. [40], position of *V. priscus* taken from ref. [41]). Red branches indicate taxa with obvious, consistently pigmented cutting edges. Black branches indicate those with no obvious pigmentation. Blue branches indicate taxa with occasional or inconsistently pigmented teeth. Orange branch indicates unknown level of pigmentation due to fossilization. Scale bars, 1 mm. **b**, Isolated tooth crown of the extinct *V. priscus* (SAMP 54739) showing no signs of pigmentation on the distal serrations. **c**, A second isolated crown assigned to *V. priscus* (SAMP 54739) showing no evidence of pigmentation.

extend out from the enamel–dentine junction[7,22] (Fig. 6c,d). However, this enamel structure differed along the serrations: crystallites within the outer half of the serration enamel form clusters that spiral towards the outer enamel surface (Fig. 6g,h). This enamel type is reminiscent of wavy enamel found in hadrosaurs, a herbivorous group of dinosaurs with grinding teeth[7,22]. Wavy enamel forms characteristic wave patterns

under cross-polarized light[23,24] which are also present along tyrannosaurid serrations in thin section (Fig. 6f and Supplementary Fig. 17). By comparison, this wavy pattern was largely absent from the smaller teeth of dromaeosaurids, where the enamel was usually of a simpler, parallel crystallite type across the whole tooth (Supplementary Fig. 16).

To quantify the variation in enamel structure and texture along tyrannosaurid tooth crowns, we performed synchrotron-based X-ray microdiffraction (S-μXRD) mapping of thinned tooth samples. Previous studies of fossil reptile enamel using S-μXRD have studied powdered theropod dinosaur teeth or small transect maps of fossil and modern crocodylian teeth[8,25,26] and thus were unable to provide spatial information along a tooth crown. In contrast, our mapping of apatite crystallite orientations as well as the texture (variance in orientation) in situ along and across entire tyrannosaurid serrations allowed us to quantify structural variation along the section, a method that has previously been used on human teeth[27] (Supplementary Figs. 18–20). In tyrannosaurids, the dentine was comparatively poorly textured, with no obvious preferred crystallite orientations (Fig. 6j,k). Crystallites within the serration enamel, by comparison, showed one to three principal orientations, depending on the region of enamel. Along the serrations, the inner layers possessed a single principal apatite crystal orientation with crystal *c* axes largely parallel to the enamel–dentine junction but had a bidirectional crystallite orientation which increasingly diverges towards the serration surface (Fig. 4i,j). In horizontal section, the enamel textures on- and off-serration were significantly different (Supplementary Fig. 20 and Supplementary Table 3). The columnar enamel observed by SEM along most of the crown retains three principal orientations at acute angles to one another, reflecting the herringbone arrangements of the enamel crystallites within each column (Fig. 6e,k). In contrast, serration enamel possessed more textured, divergent, bidirectional orientations associated with the wavy enamel we observed under SEM (Fig. 6h–k).

### Mechanical testing of reptilian enamels

Nanomechanical testing of cut and polished *V. komodoensis* and *A. mississippiensis* teeth revealed that enamel hardness and elastic moduli of extant reptile teeth were comparable to various mammalian enamel types[14] (Supplementary Figs. 21 and 22). Reliably testing the iron coatings of the serration enamel in *V. komodoensis* proved difficult because it was <1 μm thick in polished sections; however, we were able to directly test thicker iron-enriched regions in an *A. mississippiensis* tooth (Supplementary Fig. 22). The indentation hardness of these iron-enriched enamel regions ($\bar{x}$ = 3.70 GPa) was slightly (9%) but significantly higher than that of underlying enamel ($\bar{x}$ = 3.36 GPa; $t$ = 2.89, $P$ = 0.0054). However, the indentation moduli (a measure of stiffness) did not differ greatly across the enamel (Fig. 4j, Supplementary Fig. 22 and Supplementary Table 4), which is consistent with measurements in some iron-pigmented mammal teeth[14]. Tyrannosaurid enamel and dentine were typically twice as hard as analogous regions in the extant *V. komodoensis* and *A. mississippiensis* teeth, probably reflecting the effects of fossilization (Supplementary Figs. 23 and 24).

### Discussion

Despite their use as analogues for theropod dinosaur feeding behaviour[9,11,28], our study demonstrates a striking and previously overlooked predatory adaptation in the Komodo dragon, *V. komodoensis*: iron-enriched, protective layers along their tooth serrations and tips. Iron sequestration is found in the dental enamel of specialized mammals[14,15,29,30]; however, the ability to sequester iron into a discrete coating along the cutting edges of a tooth has never been observed, let alone in a reptile. Furthermore, unlike in pigmented rodent teeth where mixed-phase iron oxides are incorporated into the intergranular spaces within enamel, iron appears to be concentrated into a distinct coating of ferrihydrite which is bonded to the underlying crystalline enamel in *V. komodoensis* (Fig. 2 and Supplementary Fig. 9). This was

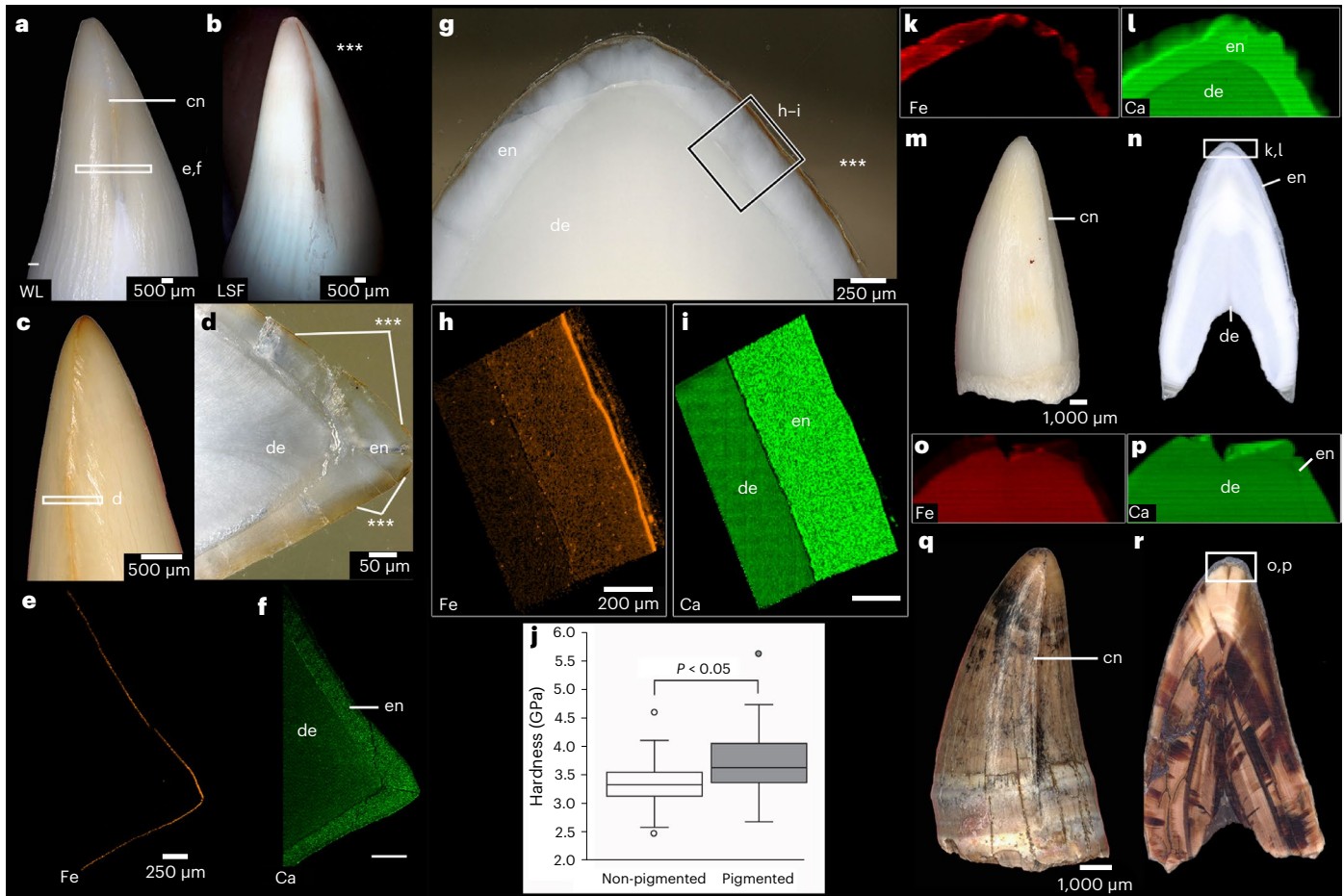

**Fig. 4 | Iron sequestration and pigmentation in crocodylian teeth. a,b**, Close-up of a posterior tooth of *Tomistoma schlegelii* under WL (**a**) and LSF (**b**), revealing a distinctive fluorescence pattern along the carina (asterisks). **c**, Close-up of the tooth tip of an anterior *T. schlegelii* tooth showing an orange carina under WL. **d**, Polished horizontal thick section through the carina of the tooth in **c** showing orange pigmentation (asterisks). **e,f**, LA-ICP-MS map of iron (**e**) and calcium (**f**), taken over a horizontal section through a carina from the tooth in **a**. **g**, Polished thick section of a caniniform tooth of *A. mississippiensis* cut parallel to the carinae, showing orange pigmentation along the outer enamel region (asterisks). **h,i**, LA-ICP-MS maps of iron (**h**) and calcium (**i**), from the enamel and dentine of the tooth in **g**. **j**, Welch test of hardness of pigmented and non-pigmented enamel in the same tooth measured using nano-indentation (*n* = 38 (non-pigmented) and 36 (pigmented) indents in one tooth). Central lines represent medians (3.33, 3.63), upper and lower bounds of boxes represent lower quartiles (3.14, 3.38) and upper quartiles (3.55, 4.06), minima (2.47, 2.68) and maxima (4.6, 5.63) for non-pigmented and pigmented enamel, respectively. *P* value (two-tailed) was 0.0058. **k,l**, S-μXRF maps of iron (**k**) and calcium (**l**) along the tip of a tooth section from a *C. porosus* tooth cut along the carina. **m**, Mesial view of the tooth sectioned and mapped in **k** and **l. n**, Section used in **k** and **l**, showing region of interest. **o,p**, S-μXRF maps of iron (**o**) and calcium (**p**) taken from a fossilized crocodylian tooth from Dinosaur Provinical Park (Canada) (UALVP 60550). **q**, Mesial view of UALVP 60550 before sectioning showing no evidence of pigmentation along the carina. **r**, Section used in **o** and **p** showing region of interest. In all elemental maps, brighter colours indicate higher counts. cn, carina.

unexpected, given that *V. komodoensis* possesses only 15–20 μm of enamel and replaces its teeth rapidly[12].

Iron-pigmented enamel has been studied in beavers[15], shrews[14], some fish[31] and salamanders[32,33] but its function remains unclear[14]. In some cases, iron sequestration hardens the enamel[15]; in others the mechanical properties of pigmented and unpigmented regions are statistically indistinguishable[14]. The iron-enriched regions in *A. mississippiensis* teeth provided us with thick enough material for nanomechanical testing and we determined that the pigmented regions exhibited slightly higher indentation hardness compared with unpigmented enamel. This local increase in hardness indicates that iron sequestration may serve to make the outer enamel layers along the cutting edges more wear-resistant compared with other regions of the tooth. Inferring a similar function to the serrated teeth of *V. komodoensis* is more difficult, given that the iron coating (and the enamel in general) is so thin. However, the coating covers each serration and is most pronounced in extant varanid species with highly serrated cutting edges, further indicating a role in supporting ziphodont tooth

function. Consistent with this, the serrations also show different mechanical wear compared with other regions of the tooth crown: most of a worn crown of *V. komodoensis* can show spalling of enamel and exposure of the underlying dentine, whereas the serration enamel appears to wear gradually (Supplementary Fig. 25). This thin iron coating must therefore be enough to retain a sharp cutting edge on each tooth before it is replaced by a new generation, which would have occurred very frequently[12]. Other proposed functions for iron pigmentation include increased abrasion-, acid- or microfracture-resistance[14,30]. The iron layer is acid-resistant in *V. komodoensis*, based on SEM observations of acid-etched teeth (Supplementary Fig. 26), and it may therefore also protect tooth serrations from digestive acids.

Our ability to visually detect the same pigmented cutting edges in closely related species of *Varanus* suggests that iron sequestration may also be widespread in reptile teeth. We therefore predicted to see this feature in the large ziphodont teeth of closely related extinct taxa as well (for example, *V.* ('*Megalania*') *priscus*). However, our broader comparisons show that iron sequestration is neither a presence-or-absence

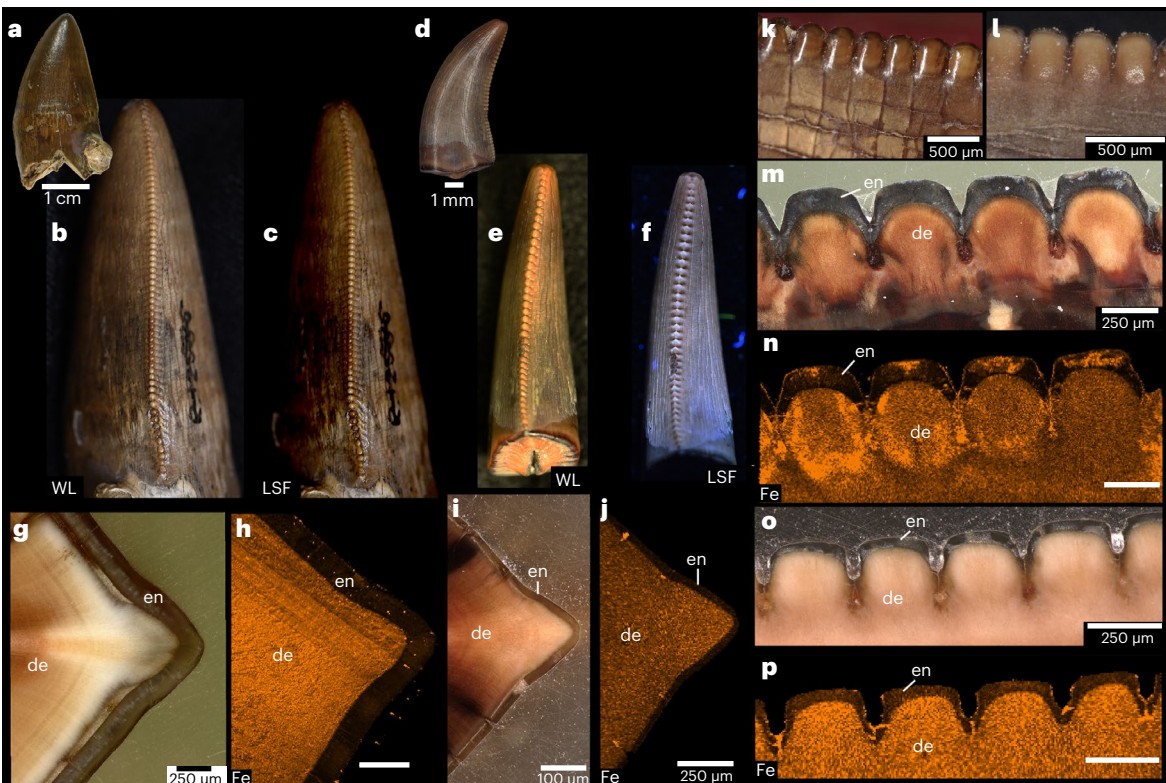

**Fig. 5 | Iron distribution within fossilized theropod tooth serrations.**
**a**, Labiolingual view of a tooth of the tyrannosaurid *Albertosaurus* (NHMUK R12599). **b,c**, Distal view of NHMUK R12599 under WL (**b**) and LSF imaging (**c**), showing a lack of differential fluorescence between the serrations or the rest of the tooth. **d**, Labiolingual view of a dromaeosaurid tooth (UALVP 61165). **e,f**, Distal view of a dromaeosaurid tooth (UALVP 61165) under WL (**e**) and LSF imaging (**f**), showing no fluorescence difference on- or off-serration. **g**, Polished horizontal thick section through a distal serration of a tyrannosaurid tooth (UALVP 60554). **h**, LA-ICP-MS map of iron along the distal serration in g showing no evidence of iron sequestration along the serration enamel.

**i**, Polished horizontal thick section through a distal serration of UALVP 61165. **j**, LA-ICP-MS map of iron along the same region as in **g**. **k**, Close-up view of serration enamel in a tyrannosaurid tooth (UALVP 60553). **l**, Serration enamel in a dromaeosaurid tooth (UALVP 61165). **m**, Polished thick section through distal serrations of a tyrannosaurid tooth (UALVP 60555). **n**, LA-ICP-MS map of iron along the serrations in UALVP 60555. **o**, Polished thick section through the distal serrations of a dromaeosaurid tooth (UALVP 61165). **p**, LA-ICP-MS map of iron along the serrations in UALVP 61165. In all elemental maps, brighter colours indicate higher counts.

phenomenon in extant reptiles nor is it easily detectable in fossil reptile teeth. Even where pigmentation was absent from the cutting edges in extant reptile teeth under white light, fluorescence imaging techniques revealed that even extant crocodylians can possess iron-enriched outer enamel layers, particularly along the cutting edges. This suggests that many reptiles may have iron-enriched enamel but that only some species have evolved prominent iron coatings along specific parts of their tooth crowns, presumably as feeding adaptations. Under high enough concentrations, the iron layers are visible under white light, as is the case in *V. komodoensis* and some of its ziphodont relatives.

Given that extant archosaurs can sequester iron within their enamel and that *V. komodoensis*, the largest extant ziphodont reptile, possessed iron-coated tooth serrations, we predicted that we may be able to detect similar iron coatings in extinct ziphodont varanids and theropod dinosaurs. However, even in cases where we might expect to observe this adaptation, diagenesis may obscure any site-specific iron signal within the enamel. We did not detect any colour changes along the serrations of *V. priscus* and we did not see any consistent iron-enrichment within the serration enamel of theropod dinosaurs using any technique.

Despite the lack of evidence for iron coatings along theropod tooth serrations, our sample of tyrannosaurid teeth demonstrated a second surprising adaptation within ziphodont reptile teeth. SEM imaging and S-µXRD mapping consistently showed microstructural differences between the serration and non-serration enamel in tyrannosaurid teeth. Tyrannosaurids possessed a form of wavy enamel along

the tooth serrations that is elsewhere only found on the grinding teeth of herbivorous ornithopod dinosaurs[7,22]. Unlike in ornithopods, wavy enamel is restricted to the serrations in tyrannosaurid teeth, rather than along the entire crown (Fig. 6). This specialization of the serration enamel in tyrannosaurids suggests a role in supporting the cutting edges of the teeth. In ornithopods, this staggered, sinusoidal arrangement of crystallite bundles may have hindered crack propagation and avoided larger scale fracturing of the enamel, owing to the convoluted pathways in which microfractures could form[7]. As in *V. komodoensis*, tyrannosaurid teeth frequently show evidence of enamel spalling[34], however this is not typically seen on the serrations themselves (Supplementary Fig. 25).

Alternative feeding strategies between *V. komodoensis*[9] and some tyrannosaurids[10,35] may explain these divergent strategies for maintaining serrated teeth but these differences could also reflect scaling effects between these two ziphodont taxa. Dromaeosaurid dinosaur teeth provide partial evidence for this. The dromaeosaurid teeth sampled in this study were of similar sizes to those of *V. komodoensis* and most showed comparably thin and simple enamel microstructure between the serrations and the rest of the crown (Supplementary Fig. 16). Only in a dromaeosaurid tooth with thicker enamel (40 µm or more) did we see any qualitative differences in enamel structure on- and off-serration but these were still not as pronounced as in the larger tyrannosaurid teeth (Supplementary Fig. 16k–o). Despite having relatively thin enamel coatings, it is possible that only the larger teeth of theropod dinosaurs

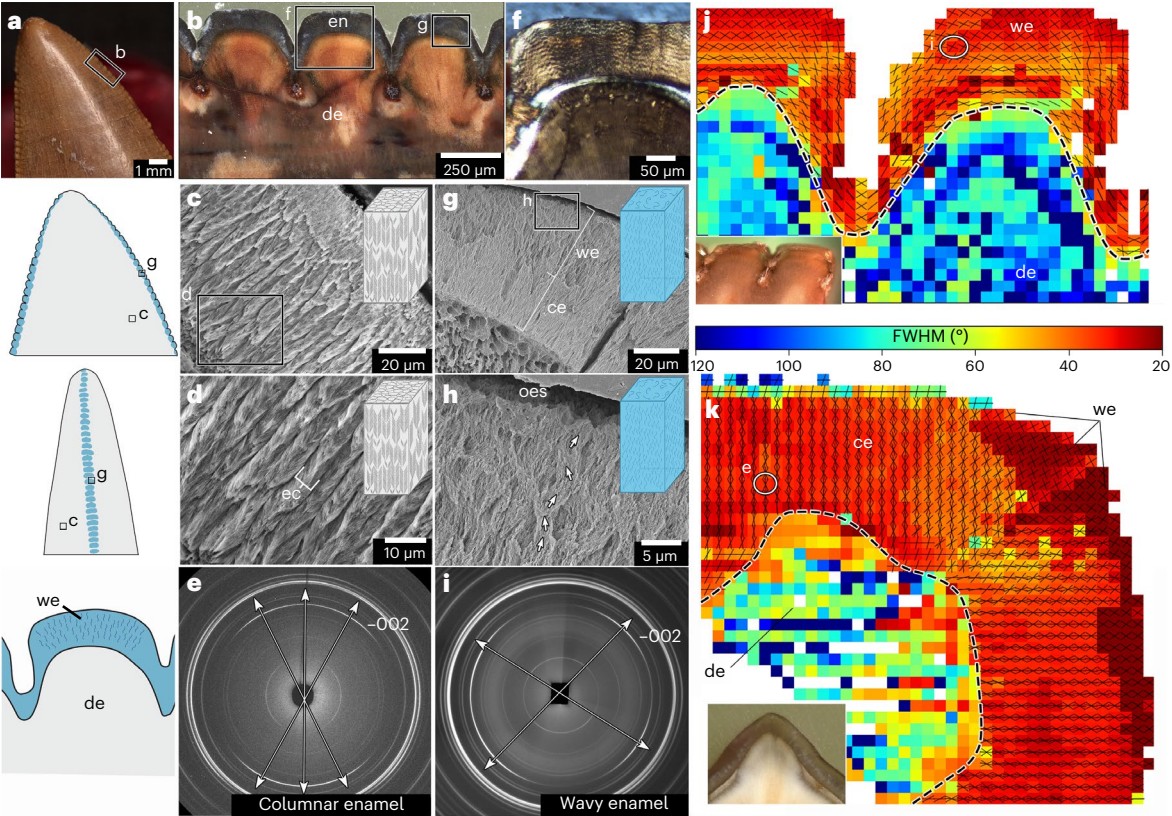

**Fig. 6 | Structural specializations of the serration enamel in tyrannosaurid teeth. a**, Close-up image of an isolated tyrannosaurid tooth (UALVP 60556), with illustrations of the distributions of columnar (grey) and wavy (blue) enamel based on structural analyses. **b**, Polished thick section through the distal serrations of UALVP 60555 showing positions of sections in subsequent analyses. **c**, SEM image of columnar enamel in a tyrannosaurid tooth in horizontal section (UALVP 60556). **d**, Higher magnification SEM image showing herringbone stacks of enamel crystallites within each column. **e**, S-µXRD-generated diffraction pattern taken through a region of columnar enamel in a tyrannosaurid tooth (UALVP 60554), showing three principal enamel crystallite orientations in the 002 reflection. **f**, Cross-polarized light image of serration enamel in a thin section of a tyrannosaurid tooth (UALVP 60398) showing wavy pattern in enamel. **g**, SEM image of enamel along a serration of a tyrannosaurid tooth in longitudinal section (UALVP 60557). **h**, Higher magnification image of the outer enamel

surface showing the spiralled arrangements of serration enamel. **i**, S-µXRD-generated diffraction pattern taken through a region of wavy enamel in a tyrannosaurid tooth, showing two principal enamel crystallite orientations in the 002 reflection and a higher divergence angle between them compared with the columnar enamel (**e**). **j**, Composite apatite crystal texture map of two serrations in longitudinal section (UALVP 53472) derived from diffraction patterns. **k**, Composite texture map of enamel crystallites along a single serration in horizontal section (UALVP 60554). Lines within each pixel in **j** and **k** indicate principal enamel crystallite orientations. Hotter colours indicate more highly textured regions (lower full-width half maxima). Wavy enamel manifests as larger divergences between principal orientations and more textured regions along the serration. ce, columnar enamel; ec, enamel column; FWHM, full-width at half-maximum; oes, outer enamel surface; we, wavy enamel.

possess enough enamel over which the microstructure can influence the mechanical wear of the serrations. Conversely, the enamel is probably too thin in extant ziphodont lizards to influence tooth wear on a similar scale to tyrannosaurids. The thin, structurally simple enamel in *V. komodoensis* probably plays a limited role in the cutting function of each tooth and natural selection may have alternatively favoured individuals with more elaborate iron coatings to mitigate wear along their rapidly replaced teeth. If theropod dinosaurs did similarly incorporate iron into their ziphodont teeth, then we may expect to find clearer evidence in smaller species or individuals with comparably thin enamel. Identifying these iron coatings in fossil reptile teeth, however, remains a challenge; we did not see any evidence for iron sequestration, even in small dromaeosaurid teeth.

Our results highlight two unexpected and disparate evolutionary adaptations in ziphodont reptiles. On the one hand, *V. komodoensis* evolved prominent iron coatings along the tooth tips and serrations to reinforce their cutting teeth. By comparison, some theropod dinosaurs developed structural adaptations in enamel along their tooth serrations. In both cases, the cutting edges of the teeth in these ziphodont reptiles are either structurally or chemically modified. Despite having

remarkably thin enamel, Komodo dragon and tyrannosaurid teeth both highlight the reptilian capacity to modify specific regions of the dental hard tissues to maintain sharp cutting edges. Whereas structurally and chemically complex enamel are traditionally associated with derived mammalian groups[14,15,30], it is evident that similar adaptations lie hidden within the thin veneer of enamel of carnivorous reptiles and probably contributed to their success as apex predators.

## Methods

### Light microscopy

Imaging of Natural History Museum London (NHMUK), Museum of Life Sciences (MoLS) and South Australian Museum (SAM) specimens was done using a Nikon D810 DSLR camera or a Keyence VHX 7000 digital microscope (King's College, London). American Museum of Natural History (AMNH) specimens were imaged using a Leica digital dissecting microscope.

### Sample preparation for analyses

Samples were all first embedded in epoxy resin (EpoThin2, Beuhler) and placed under vacuum. Samples were then sectioned along planes

of interest using an Isomet 1000 low-speed saw (Buehler) and polished using a LaboForce (Struers) grinding and polishing wheel, using P1200, P2000 and P4000 grinding papers and aluminium oxide polishing powders on cloth plates. Samples used for SEM were etched for 15–30 s using 1 M hydrochloric acid, placed in distilled water for several minutes, then mounted to SEM stubs. Samples prepared for S-µXRF and S-µXRD analyses were embedded in the same resin and cut using the Isomet saw to thicknesses of ~500 µm, then glued to blank epoxy resin pucks for mounting on to the LaboForce polishing machine. Samples were then ground to thicknesses of <300 µm using the same grinding papers and polishing powders before being removed from the resin pucks using a dental drill.

## Scanning electron microscopy–energy dispersive spectroscopy

SEM imaging was conducted in two facilities: Centre for Ultrastructural Imaging (CUI) at King's College London and Harvey Flowers Electron Microscopy Suite, Imperial College London. Samples were coated with 6–10 nm of gold and imaged using a JEOL JCM-7000 or a JEOL JSM-7800 F Prime SEM (CUI). These samples were imaged under high-vacuum setting using a 15 kV beam acceleration voltage on a JCM-7000 SEM (JEOL). SEM–EDS analysis was performed on a Sigma 300 SEM (Zeiss) equipped with an XFlash 6–60 EDS detector (Bruker), using 10 kV beam accelerating voltage (Harvey Flowers Electron Microscopy Suite).

STEM cross-sections were prepared from the SEM sample using a Helios 5 CX (ThermoFisher Scientific) with a Ga ion beam following standard milling procedures. Samples were milled to a thickness of 70–100 nm.

## Electron microscopy–energy dispersal spectroscopy

TEM cross-sections were prepared from the SEM sample using a ThermoFisher Scientific (TFS) Helios 5 CX FIB-SEM with a Ga ion beam following standard milling procedures. Samples were milled to electron transparency with a thickness of 70–100 nm.

STEM imaging and EDS analysis was carried out on a TFS Spectra 300 TEM equipped with a monochromator and probe corrector. High-angle annular dark-field (HAADF) images were collected on a Fischione HAADF detector where the contrast in the image is roughly proportional to the atomic number squared ($Z^2$) and the mass-thickness of the sample. EDS was acquired at 300 kV from a Dual-X windowless EDX detector with a solid angle of 1.8 srad. TFS Velox software (ThermoFisher Scientific) was used to process the data.

## Laser-stimulated fluorescence

LSF of fossil theropod teeth was performed with a 405 nm diode laser projected through a lens as a line following a previously published methodology[13]. The line was scanned over the specimen in a dark room and a Nikon D850 camera with a 450 nm blocking filter recorded the image. LSF imaging of crocodylian teeth was conducted using the same 405 nm diode laser projected as a cone and imaged using the Keyence VHX 7000 microscope with a 505 nm blocking filter. Images were colour equalized in Photoshop (Adobe, v.25.5), applied equally to the entire image.

## Synchrotron radiation X-ray microfluorescence spectroscopy

S-µXRF analyses were undertaken at the Diamond Light Source (Oxford Harwell Campus) beamline B16 (experiments M22284-2019 and MM26050-2020) and European Synchrotron Radiation Facility (Grenoble, France) beamlines BM28 (experiments 28-01-1286 and LS-3093) and ID21 (experiment LS-3074). All experiments were conducted using a monochromatic beam. Data acquisition parameters for each experiment and sample used in this study are detailed in Supplementary Table 2.

For data processing, raw XRF data were imported into PyMCA (v.5.6.3), calibrated to the beam energy, batch fitted and normalized to incident beam intensity for each scan. Maps were then generated in Matplotlib[36] in Python (v.3.9.7).

## Synchrotron radiation X-ray microdiffraction mapping

Synchrotron X-ray microdiffraction (S-µXRD) mapping measurements were conducted on the B16 beamline[37] at Diamond Light Source and the BM28 beamline at the European Synchrotron Radiation Facility. An incident X-ray energy of 15.5 keV was used, equivalent to a wavelength of 0.799 Å, with a beam size of 24 × 19 µm² (horizontal × vertical). Specimens were mounted in a transmission geometry normal to the impinging X-rays onto a travelling x–y sample stage to enable measurements in two orthogonal directions perpendicular to the X-ray beam. A single S-µXRD measurement had an exposure time of 20 s and was collected using a two-dimensional (2D) area detector (Image Star 9000, Photonic Science) with a 3,056 × 3,056 pixel resolution (31 × 31 µm² optical pixel size), placed 144 mm behind the sample to give a q range of 0.15–3.46 Å⁻¹ (where $q = 4\pi/\lambda \sin\theta$). For each specimen, raster mapping measurements were collected at spatial increments of 20 µm in the x–y directions, with corresponding values for beam transmission recorded throughout using a beam stop mounted photo-diode. Additional measurements were taken for the direct beam and empty sample holders to allow the diffraction patterns to be corrected for background and normalization effects. Instrument parameters including the X-ray wavelength and experimental geometry were accurately determined by calibrating to a lanthanum hexaboride (LaB6) (Sigma Aldrich) standard sample. Other experimental parameters are summarized in Supplementary Table 2.

Two-dimensional diffraction images presented in Fig. 4j,k were processed using the DAWN (v.2.27.0, 2022) software package[38,39]. Diffraction images were normalized with respect to count time, monitor intensity and transmission and background corrected. Data were then remapped to polar coordinates, that is cake remapping, to generate images of azimuthal angle (°) versus q (Å⁻¹) (x versus y) to enable 2D fitting of the (002) diffraction peak located at ~1.83 Å⁻¹. The (002) plane was used to measure the c lattice parameter as a function of location within the sample map and preferred orientation as it is normal to the c axes of the fluorapatite unit cell. Here, preferred orientation is determined from intensity variations in the (002) diffraction ring with respect to the azimuth. To facilitate batch fitting of relatively large diffraction data of highly textured specimens, a machine learning based approach was used to classify data according to preferred orientation (Supplementary Figs. 18 and 19), which demonstrated greater variance in the diffraction images compared with the peak positions in q space. Remapped data were sectioned about the (002) diffraction peak and averaged along the azimuthal axis to produce one-dimensional plots of intensity versus the azimuthal angle (Supplementary Fig. 18a) representing the preferred orientation of constituent crystallite populations. Principal component analysis (PCA) was applied to the orientation data (Supplementary Fig. 18b), following normalization and mean centring, to aid in texture classification as a function of position within the specimen. Orientation data were grouped according to their similarity using k-means clustering applied to the PCA data (Supplementary Fig. 18c), with ~40–50 unique clusters required to classify each S-µXRD map.

To improve data clarity and aid in the identification of orientation peaks, data were reconstructed from their principal components to remove components which explained little variance (only applied for peak identification purposes) and data were subsequently averaged in each cluster. For each averaged data cluster, peak positions and peak widths were determined from the second derivative of the intensity of the orientation data (Supplementary Fig. 18d). Troughs in the second derivative corresponded to underlying peak positions while peak widths were estimated from trough widths, where the width of a second derivative trough is approximately one-third of a peak width in the intensity domain. Values for peak positions and widths were used as parameters to initialize 2D pseudo-Voigt peak models on a plane background to fit the 002 peaks in the original (sectioned) remapped diffraction images for each cluster (Supplementary

Fig. 18e). The quality of data fitting was determined by least-squares methods where the goodness-of-fit increased as the chi-square value approached unity. The computational analysis for the processing and fitting pipeline was performed using Python programming software (v.3.9.12, 2022). The 2D peak positions in the $x$ and $y$ axis of the diffraction images yielded the direction of preferred orientation and the (002)-peak position in $q$ space, respectively, for each constituent crystallite population. The positions of the (002) peaks were used to calculate the corresponding unit cell $c$ axis lattice parameters. A hexagonal unit cell was assumed for fluorapatite, where the relationship between the $d$ spacing ($d = 2\pi/q$), Miller indices for a given Bragg peak and lattice parameters are given as:

$$\frac{1}{d^2} = \frac{4}{3}\left(\frac{h^2 + hk + k^2}{a^2}\right) + \frac{l^2}{c^2} \qquad (1)$$

where $h$, $k$ and $l$ refer to the Miller indices for a particular Bragg peak while $a$ and $c$ represent the relevant lattice parameters. Substituting the (002) indices into equation (1) gives the individual $c$ lattice parameter as a function of $d$.

## Nano-indentation
A *V. komodoensis* (J94036-2) and tyrannosaurid (UALVP 60555) tooth were cut in longitudinal section to expose the mesial and distal serrations. These were polished using a LaboForce (Struers) grinding and polishing wheel, using P1200, P2000 and P4000 grinding papers and aluminium oxide polishing powders on cloth plates. A Bruker Hysitron Ti950 with a Berkovitch indenter tip was used, equipped with a load-controlled, trapezoidal function (4,000 μN of force, 5 s loading onto the sample, 5 s sustain, 5 s release). Analyses were conducted on regions of interest along mesial and distal serrations of each tooth. Owing to the order of magnitude difference in size and thickness of enamel between the *V. komodoensis* and tyrannosaurid teeth, different spacings between indents for both samples were used. For *V. komodoensis*, successive indents were 10 μm apart to ensure that several measurements were taken in the thin enamel. For the tyrannosaurid tooth, indents were spaced 15 μm apart.

Two shed teeth from an *A. mississippiensis* individual from the Crocodiles of the World Zoo (Brize Norton, UK) were also sectioned longitudinally along the carinae. 'Tooth 1' was a posterior, bulbous tooth, whereas 'tooth 2' was a conical anterior tooth. These teeth were polished following the same protocol and a series of transects of indentations were taken through the enamel of each tooth (Supplementary Fig. 22). Indents were spaced 15 μm apart using the same loading function as with the other tooth samples (4,000 μN of force, 5 s loading onto the sample, 5 s sustain, 5 s release). Following the LA-ICP-MS experiments on 'tooth 2' (Fig. 4g–j), we conducted a second indentation experiment on the outer enamel layers of the tooth to ensure that we indented the iron-rich, slightly pigmented regions in the outer 30–40 μm. We conducted a series of indentations along the pigmented and unpigmented enamel regions just below the two regions we sampled for LA-ICP-MS (Supplementary Fig. 22e,o–q). We recorded a total of 36 indents within the pigmented enamel regions and 38 indents in adjacent, underlying enamel regions. We then compared the indentation hardness and elastic moduli between the unpigmented and pigmented enamel regions using a two-sample Welch test and a pooled variance $t$-test on the hardness and elastic modulus data, respectively. Effect size was calculated using Cohen's $d$ (Supplementary Table 4).

## Iron-edge X-ray absorption near-edge structure
Fe K-edge XANES measurements were performed at the I-18 beamline (Diamond Light Source, experiment SP35162-1) with a $2 \times 2$ μm² beam footprint. XRF mapping was first undertaken to identify Fe-rich regions of interest within the enamel of *V. komodoensis*, *C. porosus* and *C. canadensis* before point XANES measurements (three to five per sample). For XANES, the energy of the incoming beam was scanned from 6.96 to 7.25 keV with increments of 0.5 eV over the pre-edge, edge and post-edge features. Energy calibration was performed using an Fe foil. Spectra were analysed in Athena XAS Data Processing (Demeter v.0.9.26). Spectral standards for ferrihydrite ($Fe_5HO_8\cdot4H_2O$), magnetite ($Fe_3O_4$) and haematite ($Fe_2O_3$) were sourced from https://xaslib.xrayabsorption.org (International X-ray Absorption Society) and were energy-normalized using Fe foil measurements before analysis.

## Laser ablation inductively coupled mass spectrometry
An Iridia 193 nm ArF*excimer-based LA system (Teledyne Photon Machines) was used, equipped with the cobalt long-pulse ablation cell. The LA system was coupled to a ThermoFisher Scientific iCAPTQ ICP-mass spectrometer (ThermoFisher Scientific) by means of the Aerosol Rapid Introduction System. Full operational parameters for both the Iridia and iCAPTQ ICP-MS are provided in Supplementary Table 5.

Tuning of the instrument settings was performed using a NIST SRM 612 glass certified reference material (National Institute for Standards and Technology), optimizing for low laser-induced elemental fractionation by monitoring $^{238}U^+/^{232}Th^+$, oxide formation rates (<1%) through the $^{232}Th^{16}O^+/^{232}Th^+$ ratio and the sensitivity of $^{59}Co^+$, $^{115}In^+$ and $^{238}U^+$. LA-ICP-MS image was acquired in a fixed dosage mode, with a vertical and horizontal spatial resolution of 5 μm. Imaging was performed on extant crocodylian and fossil tyrannosaurid tooth samples. The selected isotopes of interest $^{24}Mg$, $^{44}Ca^{16}O$, $^{56}Fe$, $^{66}Zn$, $^{88}Sr^{16}O$, $^{89}Y^{16}O$ and $^{138}Ba^{16}O$ were chosen to maximize sensitivity while minimizing isobaric/polyatomic interferences and increasing signal-to-noise ratio. dynamic reaction collision mode with oxygen as the reaction gas was used in the collision reaction cell to reduce the contribution of polyatomic interferences on imaging for all isotopes. The reactive oxygen species measured are proxies for the elements of interest (Ca, Sr, Y and Ba) and are therefore labelled in the LA-ICP-MS maps accordingly. To correct for instrumental drift, a series of NIST 612 standard ablation scans were performed before and after experimental samples.

ICP-MS and positional data were reconstructed to generate elemental images using the hierarchical data format (HDF)-based image processing software (HDIP, Teledyne Photon Machines). A bespoke in-house pipeline, written in Python (v.4), was used to analyse the reconstructed data and produce comparable elemental images. The pipeline consisted of removing negative values, attributed to instrumental noise and replacing with zeros. A mask was applied to each image removing noise and reconstruction artefacts surrounding the samples. All data are presented on the same intensity scale for comparability (95% percentile).

We then used the Keyence digital microscope to generate 2D profiles of the ablated regions and determined that the laser removed 5.4 times more dentine than enamel in the regions of interest, due to material differences between enamel and dentine in the extant crocodylian teeth. A correction factor was applied to the LA-ICP-MS data to account for this difference and generated corrected relative counts for the elements of interest (Supplementary Fig. 8).

*V. komodoensis* serrations were imaged using LA-ICP-TOF (time-of-flight)-MS using a 193 nm Iridia (Teledyne) coupled with a Vitesse (Nu Instruments). The Iridia was equipped with the cobalt short-pulse ablation cell achieving a single-shot washout of <2 ms. Laser parameters of 500 Hz (repetition rate), dosage of ten shots and a 4 μm spot size were used for imaging. Data were processed using Nu Quant software, HDIP and Python.

## Statistics and reproducibility
Electron microscope, S-μXRF and S-μXRD, nano-indentation and LA-MS experiments were all conducted once on different regions of interest on individual teeth. Several of the teeth were used for several experiments to verify elemental and structural results from different methodologies (Supplementary Information).

## Animal ethics

All museum specimens are housed in publicly accessible institutions. The fluid-preserved *V. komodoensis* from the Zoological Society of London in Fig. 1 was made available for study to A.R.H.L. following medical euthanasia at the London Zoo, the reasons for which were unrelated to this study.

## Reporting summary

Further information on research design is available in the Nature Portfolio Reporting Summary linked to this article.

## Data availability

All data, measurements and images are either in the manuscript or Supplementary Information. Raw synchrotron XRF and XRD datasets are made publicly available through the relevant synchrotron facility 3 years after each experiment under a CC-BY-4 license (see Supplementary Table 2 for experiment numbers).

## Code availability

All scripts used to analyse the synchrotron X-ray diffraction data are provided at GitHub https://github.com/Freeman163/LeBlanc.-et-al.-2024.-Nat.-Ecol.-Evol.

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

## Acknowledgements

For assistance with specimen acquisition and loans, we thank P. J. Currie, H. Gibbins, C. Coy (University of Alberta) and L. Rowden (Zoological Society of London, London Zoo). We also thank J. Streicher and S. Maidment (NHMUK), G. Sales and S. Ryder (MoLS), D. Kizirian and L. Vonnahme (AMNH), M. S. Y. Lee, M. Hutchinson and M. -A. Binnie (SAM) for collections access. E. Radvar (King's College London), P. Machado and L. Allison (CUI) assisted with SEM imaging and we thank the London Metallomics Facility for LA-ICP-MS and Nu Instruments and Teledyne for access to ICP-TOF-MS (Vitesse and Iridia). P. Shaw (Nu Instruments) and G. Foster, J. A. Milton (University of Southampton) and T. Stewart (King's College London) assisted with the Vitesse imaging. We are grateful to the synchrotron beamline staff E. Villalobos, H. Castillo Michel (ESRF-ID21), L. Bouchenoire (ESRF-BM28) and O. Fox (DLS-B16) for their assistance with the synchrotron experiments. O.A. was supported by the Natural Sciences and Engineering Research Council of Canada (grant no. 2017-05862). A.R.H.L. was supported by a Crispian Scully Research Award from the Oral and Dental Research Trust and the British Society of Dental Research. F. Giuliani was supported by the Engineering and Physical Sciences Research Council (grant no. EP/V007661/1). M. Pittman was supported by The School of Life Sciences of The Chinese University of Hong Kong. This study is part of a project that received funding from the European Research Council under the European Union's Horizon 2020 research and innovation programme Horizon (grant agreement no. 894331 to A.R.H.L. and O.A.). Part of this study also received funding from the European Research Council's Horizon 2020 programme (grant agreement no. 851705 to D.L.).

## Author contributions

A.R.H.L. and O.A. conceived the project. A.R.H.L., A.P.M., S. Sirovica, M.A-.J. and O.A. collected synchrotron diffraction and fluorescence data. A.P.M. and S. Sirovica. processed fluorescence and diffraction data. A.R.H.L., S.W., F.G. and C.M.M. collected electron microscopy data. M.P. and T.G.K. collected LSF data. A.R.H.L. and D.C.D. collected data on pigmented enamel in lizards. A.P.M. collected laser ablation mass spectrometry data. D.C.D., C.C., C.S., S. Spiro, B.T., J.C. and D.L. provided specimens and data for extant reptile teeth. A.R.H.L. collected the histology data. A.R.H.L. and D.L. conducted nano-indentation experiments. All authors assisted in interpreting the data and edited previous drafts of the manuscript.

## Competing interests

The authors declare no competing interests.

## Additional information

**Correspondence and requests for materials** should be addressed to Aaron R. H. LeBlanc.

¹Centre for Oral, Clinical & Translational Sciences, King's College London, London, UK. ²Centre for Oral Bioengineering, Queen Mary University of London, London, UK. ³School of Dentistry, University of Leeds, Leeds, UK. ⁴Evolutionary Biomechanics Laboratory, Department of Bioengineering, Imperial College London, London, UK. ⁵Department of Natural Sciences, Daemen University, Amherst, NY, USA. ⁶School of Science Engineering and Technology, University of the Sunshine Coast, Sippy Downs, Queensland, Australia. ⁷Department of Materials, Imperial College London, London, UK. ⁸School of Life Sciences, The Chinese University of Hong Kong, Shatin, Hong Kong SAR, China. ⁹Foundation for Scientific Advancement, Sierra Vista, AZ, USA. ¹⁰Crocodiles of the World, Brize Norton, UK. ¹¹Zoological Society of London, Regent's Park, London, UK. ¹²Wildlife Health Services, Zoological Society of London, Regent's Park, London, UK. ¹³Faculty of Medicine and Dentistry, University of Alberta, Edmonton, Alberta, Canada. ✉e-mail: aaron.leblanc@kcl.ac.uk

# Reporting Summary

## Statistics

For all statistical analyses, confirm that the following items are present in the figure legend, table legend, main text, or Methods section.

| n/a | Confirmed | |
|---|---|---|
| ☐ | ☒ | The exact sample size ($n$) for each experimental group/condition, given as a discrete number and unit of measurement |
| ☐ | ☒ | A statement on whether measurements were taken from distinct samples or whether the same sample was measured repeatedly |
| ☐ | ☒ | The statistical test(s) used AND whether they are one- or two-sided *Only common tests should be described solely by name; describe more complex techniques in the Methods section.* |
| ☐ | ☒ | A description of all covariates tested |
| ☒ | ☐ | A description of any assumptions or corrections, such as tests of normality and adjustment for multiple comparisons |
| ☐ | ☒ | A full description of the statistical parameters including central tendency (e.g. means) or other basic estimates (e.g. regression coefficient) AND variation (e.g. standard deviation) or associated estimates of uncertainty (e.g. confidence intervals) |
| ☐ | ☒ | For null hypothesis testing, the test statistic (e.g. $F$, $t$, $r$) with confidence intervals, effect sizes, degrees of freedom and $P$ value noted *Give P values as exact values whenever suitable.* |
| ☒ | ☐ | For Bayesian analysis, information on the choice of priors and Markov chain Monte Carlo settings |
| ☒ | ☐ | For hierarchical and complex designs, identification of the appropriate level for tests and full reporting of outcomes |
| ☐ | ☒ | Estimates of effect sizes (e.g. Cohen's $d$, Pearson's $r$), indicating how they were calculated |

*Our web collection on statistics for biologists contains articles on many of the points above.*

## Software and code

Policy information about availability of computer code

| Data collection | *Provide a description of all commercial, open source and custom code used to collect the data in this study, specifying the version used OR state that no software was used.* |
|---|---|
| Data analysis | Python v.3.7.4.; PyMCA v.5.6.3.; Microsoft Excel; DAWN v.2.27.0 |

For manuscripts utilizing custom algorithms or software that are central to the research but not yet described in published literature, software must be made available to editors and reviewers. We strongly encourage code deposition in a community repository (e.g. GitHub). See the Nature Portfolio guidelines for submitting code & software for further information.

## Data

Policy information about availability of data

All manuscripts must include a data availability statement. This statement should provide the following information, where applicable:

- Accession codes, unique identifiers, or web links for publicly available datasets
- A description of any restrictions on data availability
- For clinical datasets or third party data, please ensure that the statement adheres to our policy

All data, measurements, and images are either in the manuscript, extended data, or supplementary information. For raw synchrotron XRF and XRD datasets, these are made publicly available through the relevant synchrotron facility three years after each experiment under a CC-BY-4 license.

# Research involving human participants, their data, or biological material

Policy information about studies with [human participants or human data](). See also policy information about [sex, gender (identity/presentation), and sexual orientation]() and [race, ethnicity and racism]().

| | |
|---|---|
| Reporting on sex and gender | N/A |
| Reporting on race, ethnicity, or other socially relevant groupings | N/A |
| Population characteristics | N/A |
| Recruitment | N/A |
| Ethics oversight | N/A |

Note that full information on the approval of the study protocol must also be provided in the manuscript.

# Field-specific reporting

Please select the one below that is the best fit for your research. If you are not sure, read the appropriate sections before making your selection.

☐ Life sciences          ☐ Behavioural & social sciences          ☒ Ecological, evolutionary & environmental sciences

For a reference copy of the document with all sections, see [nature.com/documents/nr-reporting-summary-flat.pdf]()

# Ecological, evolutionary & environmental sciences study design

All studies must disclose on these points even when the disclosure is negative.

| | |
|---|---|
| Study description | Several teeth of the Komodo dragon (Varanus komodoensis) were subjected to destructive analyses and used to characterize the structure and chemistry of the enamel along their tooth serrations. Teeth were embedded in resin, sectioned, polished, imaged, and prepared for several analytical techniques. Scanning Electron Microscopy (SEM), Energy Dispersal Spectroscopy (EDS), Focused Ion Beam (FIB) milling, Transmission Electron Microscopy (TEM), Synchrotron-based X-Ray MicroFluorescence (S-XRF), and Laser Ablation Mass Spectrometry (LA-MS) techniques were used to examine the structure of the enamel as well as its elemental composition. Element counts are recorded in the supplementary information of the manuscript for S-XRF and LA-MS and highlight the presence of zinc- and iron-coated serrations in V. komodoensis, a feature that is usually associated with increased wear-resistance in mammals.<br><br>We then examined the teeth of four species of extant crocodylian for evidence of iron-coated cutting edges along their teeth to assess the possibility that archosaurs also possess this unusual adaptation. For scouting this, we used a non-destructive technique (Laser-Stimulated Fluorescence-LSF) to determine that several species of extant crocodylian had different elemental concentrations along the cutting edges of their teeth. We then subjected these teeth to S-XRF and LA-MS to determine that iron (and zinc) were similarly present along their cutting edges, similar to V. komodoensis. The same characterization techniques were also used on a small sample of fossil crocodylian teeth from the Late Cretaceous Dinosaur Provincial Park locality of southern Alberta (Canada) to determine the effect of fossilization on our ability to detect iron layers in crocodylian teeth. We then conducted the same techniques on a sample of fossil tyrannosaurid and dromaeosaurid teeth from the same locality and non-destructive LSF imaging was conducted on a larger sample of museum specimens of other theropod dinosaur teeth. These findings demonstrated that fossilized crocodylian and theropod teeth show anomalous iron and zinc signatures throughout their dental tissues, making it nearly impossible to assess wither similarly-shaped theropod teeth possessed the iron-coated serrations we identified in modern Komodo dragons.<br><br>We then conducted Synchrotron-based X-Ray MicroDiffraction (S-XRD) on a sample of the tyrannosaurid teeth to quantify changed in enamel crystal structure along the serrations compared with other regions of the teeth to determine if theropod dinosaurs used alternative strategies for increasing the wear resistance of the cutting edges of their teeth. These results, along with SEM observations, demonstrated that tyrannosaurid dinosaurs in particular evolved a complex form of enamel that is elsewhere only seen in specialized herbivorous dinosaurs with grinding teeth, suggesting that this is a microfracture-mitigating adaptation within the serration enamel of tyrannosaurids.<br><br>To assess the mechanical properties of extant and fossil reptile enamels, nanomechanical testing was undertaken at Imperial College London. |
| Research sample | Komodo dragon (Varanus komodoensis) teeth were collected from various institutions (Museum of Life Sciences, KCL; University of the Sunshine Coast, Australia; Zoological Society of London, London Zoo), some of which were then sectioned and polished for various elemental and mechanical analyses. Of these, four teeth, (MoLS X263; J94036-1; J94036-2; J94036-5), derived from two individuals, were subjected to elemental analyses. We also surveyed museum collections, as well as a recently, ethically euthanized individual of V. komodoensis for the prevalence of orange, iron-coated serrations in the species. None of these were sampled destructively.<br><br>Shed teeth from four species of extant crocodylians (Tomistoma schlegelii, Osteolaemus tetrapsis, Crocodylus porosus, and Alligator mississippiensis) were used for elemental and mechanical testing. |

| | |
|---|---|
| | Two fossil crocodylian teeth, six tyrannosaurid teeth, and three dromaeosaurid teeth were loaned to O. Addison and A. LeBlanc for destructive analyses from the University of Alberta Laboratory for Vertebrate Palaeontology (UALVP). These were sectioned at various planes. |
| Sampling strategy | Sample sizes were based on availability of material at the respective institutions for destructive and non-destructive analyses. |
| Data collection | Light and electron microscopy images were collected by A. LeBlanc in the Centre for Oral, Clinical & Translational Sciences and the Centre for Ultrastructural Imaging (KCL) using in-house Keyence VHX7000 digital microscope and JEOL JCM-7000 SEM facilities. Additional SEM imaging and EDS analyses were conducted by Siyang Wang  using a Zeiss Sigma 300 SEM at Imperial College London, while FIB milling and TEM-EDS were conducted by Catriona McGilvery at Imperial College London. STEM cross sections were prepared from the SEM sample using a ThermoFisher Scientific (TFS) Helios 5 CX with a Ga ion beam following standard milling procedures. TEM-EDS analysis was carried out at on a TFS Talos F200i equipped with an EDS detector.  EDS analysis was performed at 200kV with the electron optics optimised for EDS analysis.  TFS Velox software was used to process the data.

S-XRF, S-XRD, and Fe-XANES experiments and data collection were done by O. Addison, S. Sirovical, A. Morrell, and A. LeBlanc and the Diamond Light Source (UK) and the European Synchrotron Radiation Facility (France). Laser Ablation Inductively-Coupled Mass Spectrometry (LA-ICP-MS) was conducted on V. komodoensis and crocodylian teeth at the London Metallomics Facility. Additional laser ablation mass spectrometry of a Komodo dragon tooth using a Time-of-Flight system was conducted at the University of Southampton by Nu Instruments (credit: Phil Shaw) and appears in Supplementary Figure 3. All details are provided in the Materials and Methods, Extended Data, and Supplementary Information. Nanoindentation experiments were conducted by A. LeBlanc and D. Labonte using a Bruker Hysitron Ti950 with a Berkovitch indenter tip and the machine's proprietary software.

Laser Stimulated Fluorescence analysis was conducted by Michael Pittman and Thomas Kaye, and the images were processed using Adobe Photoshop. |
| Timing and spatial scale | Data collection for this project has been intermittent, between 2019-2024, including six synchrotron experiments at three beamlines across two facilities (UK and France), various museum collections visits in Canada, the UK, and Australia, nanomechanical testing experiments at Imperial College London, and electron microscopy at Imperial and King's Colleges. |
| Data exclusions | N/A |
| Reproducibility | Multiple teeth of Varanus komodoensis were subjected to elemental analyses to ensure repeatability of our findings. We also used multiple complementary elemental techniques (synchrotron X-ray microfluorescence; energy dispersive spectroscopy; laser ablation mass spectrometry) to ensure signals for elements of interest (e.g., iron, zinc, calcium) were consistent. We also conducted synhcrotron X-ray microfluorescence and laser ablation mass spectrometry for teeth of three species of crocodylian (Alligator mississippiensis, Crocodylus porosus, and Osteolaemus tetraspis). Where specimen-based observations were made (e.g., Supplementary information), we have recorded each specimen number and repository where these observations can be checked. For elemental and structural imaging experiments, all parameters are provided in the Materials and Methods, the Extended Data, and the Supplementary Information.

Synchrotron X-ray microdiffraction experiments on fossil tyrannosaurid teeth were conducted at two separate synchrotron facilities (Diamond Light Source, European Synchrotron Radiation Facility) and in two planes of section to ensure repeatability of our findings.

Scanning electron microscopy was conducted at two institutions (Imperial College London; Centre for Ultrastructural Imaging, KCL) |
| Randomization | No randomization processes were undertaken, due to the rarity of the extant and fossil reptile tooth samples. Samples were instead chosen based on their availability for destructive sampling. |
| Blinding | N/A |

Did the study involve field work? ☐ Yes ☒ No

# Reporting for specific materials, systems and methods

We require information from authors about some types of materials, experimental systems and methods used in many studies. Here, indicate whether each material, system or method listed is relevant to your study. If you are not sure if a list item applies to your research, read the appropriate section before selecting a response.

## Materials & experimental systems

| n/a | Involved in the study |
|---|---|
| ☒ | Antibodies |
| ☒ | Eukaryotic cell lines |
| ☐ | ☒ Palaeontology and archaeology |
| ☐ | ☒ Animals and other organisms |
| ☒ | Clinical data |
| ☒ | Dual use research of concern |
| ☒ | Plants |

## Methods

| n/a | Involved in the study |
|---|---|
| ☒ | ChIP-seq |
| ☒ | Flow cytometry |
| ☒ | MRI-based neuroimaging |

# Palaeontology and Archaeology

| | |
|---|---|
| Specimen provenance | All fossil dinosaur and crocodylian teeth used in this study were found in Dinosaur Provincial Park and are accessioned in the University of Alberta Laboratory for Vertebrate Palaeontology (UALVP). Specimens were loaned to O. Addison and A. LeBlanc for destructive analyses under loan numbers UALVP L2020-02 and UALVP L2023-06. |
| Specimen deposition | University of Alberta, Edmonton Canada. |
| Dating methods | N/A |

☐ Tick this box to confirm that the raw and calibrated dates are available in the paper or in Supplementary Information.

| | |
|---|---|
| Ethics oversight | University of Alberta Museums approved the fossil loans and destructive sampling requests to O. Addison and A. LeBlanc (UALVP L2020-02 and UALVP L2023-06. |

Note that full information on the approval of the study protocol must also be provided in the manuscript.

# Animals and other research organisms

Policy information about studies involving animals; ARRIVE guidelines recommended for reporting animal research, and Sex and Gender in Research

| | |
|---|---|
| Laboratory animals | N/A |
| Wild animals | Shed teeth from four species of extant crocodylians were collected from the enclosures of live animals in the care of the Crocodiles of the World Conservation and Education Centre in Brize Norton, United Kingdom. Fluid-preserved Komodo dragon specimen was derived from the lower jaw of an ethically euthanized individual that was under the care of the ZSL London Zoo. The animal was not euthanized for this research. Applications for research on this individual from the London Zoo were submitted and approved by the ZSL. |
| Reporting on sex | N/A- we only had sex determination for the male V. komodoensis from the London Zoo. All remaining reptile teeth were either shed or from museum specimens for which sex was not recorded. |
| Field-collected samples | N/A |
| Ethics oversight | Loan of shed crocodylian teeth was approved by the Head of Education at Crocodiles of the World (Colin Stevenson). Loan of the lower jaw of Varanus komodoensis from the London Zoo was approved by Lewis Rowden (Zoo Research Officer, Zoological Society of London). This individual had to be ethically euthanized by Zoo veterinarians for unrelated reasons. |

Note that full information on the approval of the study protocol must also be provided in the manuscript.

