## [Peer Review File · Nature Ecology & Evolution]

Peer Review Information

Journal: Nature Ecology & Evolution

Manuscript Title: Iron-coated Komodo dragon teeth and the complex enamel of carnivorous reptiles

Corresponding author name(s): Aaron R. H. LeBlanc

Editorial Notes:

Reviewer Comments & Decisions:

Decision Letter, initial version:

11th September 2023

Dear Dr LeBlanc,

Your Article, "Iron-coated Komodo dragon teeth and the complex dental enamel of carnivorous reptiles" has now been seen by three reviewers. You will see from their comments copied below that while they find your work of considerable potential interest, they have raised quite substantial concerns that must be addressed. In light of these comments, we cannot accept the manuscript for publication, but would be very interested in considering a revised version that addresses these serious concerns, specifically the broadened comparative sample recommended by reviewer 2.

We hope you will find the reviewers' comments useful as you decide how to proceed. If you wish to submit a substantially revised manuscript, please bear in mind that we will be reluctant to approach the reviewers again in the absence of major revisions.

If you choose to revise your manuscript taking into account all reviewer and editor comments, please highlight all changes in the manuscript text file

* Include a "Response to reviewers" document detailing, point-by-point, how you addressed each referee comment. If no action was taken to address a point, you must provide a compelling argument. This response will be sent back to the referees along with the revised manuscript.

* If you have not done so already we suggest that you begin to revise your manuscript so that it conforms to our Article format instructions at <http://www.nature.com/natecolevol/info/final-submission>. Refer also to any guidelines provided in this letter.

* Include a revised version of any required reporting checklist. It will be available to referees (and, potentially, statisticians) to aid in their evaluation if the manuscript goes back for peer review. A

2revised checklist is essential for re-review of the paper.

[REDACTED]

If you wish to submit a suitably revised manuscript we would hope to receive it within 6 months. If you cannot send it within this time, please let us know. We will be happy to consider your revision so long as nothing similar has been accepted for publication at Nature Ecology & Evolution or published elsewhere.

Nature Ecology & Evolution is committed to improving transparency in authorship. As part of our efforts in this direction, we are now requesting that all authors identified as 'corresponding author' on published papers create and link their Open Researcher and Contributor Identifier (ORCID) with their account on the Manuscript Tracking System (MTS), prior to acceptance. This applies to primary research papers only. ORCID helps the scientific community achieve unambiguous attribution of all scholarly contributions. You can create and link your ORCID from the home page of the MTS by clicking on 'Modify my Springer Nature account'. For more information please visit please visit www.springernature.com/orcid.

Thank you for the opportunity to review your work.

[REDACTED]

Reviewer expertise:

Reviewer #1: signed report

Reviewer #2: osteohistology and tooth function

Reviewer #3: chemical analysis of fossil teeth.

Reviewers' comments:

Reviewer #1 (Remarks to the Author):

The paper of LeBlanc et al. documents in detail the dental enamel in one of the most iconic extant

2reptiles, *Varanus komodoensis*. Using different, cutting-edge, technologies, the authors found that the teeth of this lizard possess orange, iron- and zinc-enriched coatings on their tooth serrations and tips. The authors further compare and discuss the potential of this case in other extant but also extinct reptiles (as for the latter though, strangely they confine their investigations to archosaurs, with no mention of the possibility of that case in extinct anguimorph [especially varanoid] lizards [see my comments below]). The manuscript is overall well written and the advanced technological methods presented therein seem to justify the conclusions of the authors.

Some general comments:

The authors were probably unaware of the very recently published paper of Georgalis et al. (2023), where there was documentation of the pulp cavity of extant *Varanus* and *Lanthanotus* with the aid of μ CT scanning, as well as a detailed survey of the tooth serration in varanids, covering a wide range of extant taxa. I therefore consider that this publication should be mentioned.

Georgalis, G.L., B. Mennecart and K.T. Smith. 2023. First fossil record of *Varanus* (Reptilia, Squamata) from Switzerland and the earliest occurrences of the genus in Europe. *Swiss Journal of Geosciences* 116:9 (10 pp.).

Also, I suggest mentioning somewhere (perhaps in the Introduction or the Discussion) a couple of sentences about the presence of serrated teeth in fossil varanids. This has been investigated in detail in Georgalis et al. (2019) for European fossil varanids, where all such occurrences of serrated teeth from the continent were documented (a further occurrence with serrated teeth was subsequently described in the above mentioned Georgalis et al. 2023 paper). Moreover, in the same paper of Georgalis et al. (2019), there were fossil varanid teeth from the same Neogene locality in Greece that had four different kinds of tooth serration: teeth either completely serrated (both mesially and distally), completely non-serrated, serrated mesially but not distally, and serrated distally but not mesially – although the authors stated that it was unclear whether these differences constituted some kind of intraspecific variation or diagnostic features or even taphonomic alteration, I consider that a sentence about this Greek occurrence should be mentioned in the current manuscript.

Georgalis, G.L., A. Villa, M. Ivanov, D. Vasilyan and M. Delfino. 2019. Fossil amphibians and reptiles from the Neogene locality of Maramena (Greece), the most diverse European herpetofauna at the Miocene/Pliocene transition boundary. *Palaeontologia Electronica* 22.3.68:1–99.

In the section “Pigmented cutting edges in other extant reptiles” (or at least in the Discussion), I would suggest mentioning something also (what is known about the cutting edges) for *Lanthanotus* but also for *Shinisaurus* (perhaps McDowell and Bogert 1954). Also, for helodermatids, I suggest citing Bhullar and Smith (2008). Most importantly though, here, the authors “jump” directly from the extant lizards to the Cretaceous dinosaurs. It would be so good, in the intermediate, to mention what is the case with cutting edges in extinct varanoids (or even anguimorphs in general). I would definitely cite about the European Paleogene form *Palaeovaranus* (see Georgalis and Scheyer 2019; Georgalis et al. 2021). What about the situation in the North American and European *Saniwa* (see Smith 2006; Augé et al. 2022) or the European *Paranecrosaurus* (see Smith and Habersetzer 2021) and *Melanosauroides* (see Georgalis 2017)? The authors should cite these respective primary figures for these taxa and state the shape of their cutting edges. Of course, the authors claim that their primary goal is to check these results in theropods, however, I still consider that, even this basic mention of extinct squamates need to be done, especially when the primary study object of the manuscript is a squamate. I therefore strongly suggest that a couple of sentences should be inserted here about these fossil varanoid taxa, with the appropriate relevant references I indicated (so that the reader can directly compare the figures in the original papers).

Augé, M.L., A. Folie, R. Smith, A. Phélizon, P. Gigase and T. Smith. 2022. Revision of the oldest varanid, *Saniwa orsmaelensis* Dollo, 1923, from the earliest Eocene of northwest Europe. *Comptes Rendus Palevol* 21(25):511–529.

Bhullar, B-A.S. and Smith, K. T. 2008. Helodermatid lizard for the Miocene of Florida, the evolution of the dentary in Helodermatidae, and comments on dentary morphology in Varanoidea. *Journal of Herpetology* 42:286–302.

Georgalis, G.L. 2017. Necrosaurus or Palaeovaranus? Appropriate nomenclature and taxonomic content of an enigmatic fossil lizard clade (Squamata). *Annales de Paléontologie* 103:293–303.

Georgalis, G.L., A. Čerňanský and J. Klembara. 2021. Osteological atlas of new lizards from the Phosphorites du Quercy (France), based on historical, forgotten, fossil material. *Geodiversitas* 43(9):219–293.

Georgalis, G.L. and T.M. Scheyer. 2019. A new species of Palaeopython (Serpentes) and other extinct squamates from the Eocene of Dielsdorf (Zurich, Switzerland). *Swiss Journal of Geosciences* 112:383–417.

McDowell S. B. and Bogert C. M. 1954. The systematic position of *Lanthanotus* and the affinities of the anguinomorphans lizards. *Bulletin of the American Museum of Natural History* 105(1):1–142.

Smith K.T. 2006. A diverse new assemblage of late Eocene squamates (Reptilia) from the Chadron formation of North Dakota, USA. *Palaeontologia Electronica* 9(2):5A:1–44.

Smith K.T. and J. Habersetzer. 2021. The anatomy, phylogenetic relationships, and autecology of the carnivorous lizard “*Saniwa*” feisti Stritzke, 1983 from the Eocene of Messel, Germany. *Comptes Rendus Palevol* 20(23):441–506.

Finally, what about this “Agamidae indet.” and “Iguanidae indet.” from the supplementary material? There is no further information, geographic provenance, age (fossil or extant)? If extant, why the authors did not use some precisely identified (to the species level) specimen? I consider that the authors will be able to address these important issues I raised above. If so, following this moderate revision, I will be happy to recommend this manuscript for publication. I am at your disposal for any further query.

Yours sincerely
Dr. Georgios Georgalis

Reviewer #2 (Remarks to the Author):

I think this paper by LeBlanc et al. is very interesting, as are the results. It represents a detailed application of well-established methods to provide a refreshing viewpoint and test of some historical questions/hypotheses concerning tooth structure and use in reptiles. The results are detailed and often backed up by multiple lines of analytical evidence (which combine to be a truly impressive amount of supplementary data). Nonetheless, there are some areas where it can be improved, or where additional comparisons would strengthen the results, particularly as they pertain to the Discussion, which I feel is perhaps the overall weakest section.

I have several minor comments, and then two more serious concerns/comments. One relates to the structure of the figures and their relation to the flow/readability of the paper, and the other relates to a lack of discussion of several topics that have bearing on the strength of the conclusions.

4Provided these issues are addressed, I would recommend the paper be accepted.

Structure + Figures

These next points may be primarily an imposition of the sort of limits on figure number and style by the journal, but either way, I found the figures in this paper cluttered and at times frustrating to read, since they are very dense with information and are often lettered in a way that does not flow in any consistent manner (i.e. up-down, left-right, clockwise-anticlockwise, etc). They can of course be understood with some effort, but I think the text + figures + data are structured together in a way that almost actively works against the reader following the authors' arguments without having to constantly check back and re-read sections to understand what point they are trying to get across. If the information was presented in a way that was more accessible, then I think it would be much more straightforward to follow the arguments made in the text by the authors (and I'll note that I thought the text on its own was much better, so this issue is more with the figures themselves and the integration of figure content with text points).

If it is possible, I'd suggest some of the very dense figures be split into multiple figures so that the information can be presented to readers in a more step-wise fashion, rather than all at once via the current results plates. Alternatively, one could at least add in a figure or two to synthesize the information from the plates and multiple analytical lines into summary diagrams, with readers directed to the supplement to see the various more specific outputs across the various image plates (potentially moving some of the current plates to supplement to make room if figure count is too limiting).

I understand that there is a lot going on in the paper and as a result there are many moving parts that all have to be reported and displayed in some way, so if it ends up that the changes I suggested above for figures cannot be implemented, so be it. I just wanted to mention it because I found it did impact my reading and increased the time needed to parse through what is otherwise a very solid study.

Results & Interpretations

I have some concerns with the interpretations of the results and integration with the Discussion, at least insofar as they appear to ignore several sources of variation/uncertainty.

Many of these potential sources of uncertainty are acknowledged by the authors in the text, so I don't think they go undiscussed because the authors had not considered them. But I think they do need to be addressed more directly, whether through more extensive discussions and literature comparisons, or by some additional analyses, lest aspects of the otherwise very interesting conclusions of this paper be unfairly framed as a 'just-so' story. I'm sure the authors have thought about these topics, so I imagine that their overall conclusions will not be strongly impacted by including discussion of them, and their inclusion should only strengthen the overall impact of the paper.

These sources of uncertainty and broader discussion include the following:

5I) The lack of comparison to theropods that are of similar size to *Varanus*. In the Discussion, you argue that “Alternative feeding strategies between *V. komodoensis* (de-fleshing food items) and some tyrannosaurids (scoring and ingesting bone as well as the flesh) may explain these divergent strategies for maintaining serrated teeth, but these differences could also reflect scaling effects between these two ziphodonts. Despite both having relatively thin enamel coatings, only the large teeth of theropod dinosaurs possess enough enamel over which the microstructure can vary; the enamel is too thin in extant ziphodont lizards to permit structural variations on a similar scale”, but at no point do you provide comparisons to smaller theropods (such as dromaeosaurs), which would seem to be the perfect test of the two hypotheses offered above by the authors, while avoiding the various size-based uncertainties introduced by limiting comparisons to larger theropods such as tyrannosaurs.

II) The lack of analytical comparisons to other squamates, and particularly a lack of discussion of the implications of Fe coatings being only variably present in different species of *Varanus*. To better support the interpretation of these coatings as an adaptation for reinforcing cutting teeth in komodo dragons, and contrasting them against what is being done in different archosaurs, I think it is reasonably important to explain why this feature may be so variable within *Varanus* itself, since that would presumably impact the enamel properties of these other *Varanus* species outside of *komodoensis*. You do note that it may just be that at small size the visual effect isn’t noticeable, but again, some direct analytical data on this would go a long way to supporting the core argument of the conclusions.

III) The lack of direct mammalian comparisons (or other way to assess impact of tooth replacement as a component of the strategy underscoring the management of tooth wear). To be clear, their existence is mentioned, and some minor differences are noted, but other than that it is barely examined. This seems unusual to me given how it would represent a fairly direct analogue of this sort of adaptation in a distant group of amniotes. Along the same lines, tooth replacement rates and strategies of polyphyodonty vs. diphyodonty, and the role they may play in questions of tooth function, tooth wear rates, and enamel thicknesses, etc. are also almost mostly undiscussed, even though how frequently the teeth are being replaced would presumably have a factor on selective pressures if the argument is that Fe concentration in enamel is an adaptation to strengthen cutting surfaces and reduce tooth wear in taxa with thin enamel.

IV) A lack of comparison of fossil vs modern teeth of the same taxon (or close relative) to test some of the interpretations made concerning elemental concentration patterns in theropod tooth tissues being the result of fossilization. This one is less critical, and I don’t think it’s a bad conclusion at present, but I think it is something that could warrant an actual test, or at least some additional reference to literature data.

Minor Comments

- There may be a problem with figure numbering in the text, as I noticed that figure 3 was being called in-text a few times where I’m fairly sure Figure 2 was what was intended (for example, on Line 170). Similarly, Extended Data 2 is cited in text, but as far as I can tell it is not one of the files

6uploaded with the paper? Unless it is also synonymous with the Supplementary Information file (though at least some of what is mentioned in text for Extended Data 2 does not appear to be present in the Supp Info file). Perhaps everything is present but the order of things was changed at some point and not all of the old refs were caught? In any case, I'd recommend the authors go through and confirm all of their figures, supp figures, and extended data are present and being correctly cross-referenced in the text just in case this also happened elsewhere.

- Lines 258-260: this is probably better in the Discussion rather than Results, since it is mostly a speculation rather than something directly based on data. Alternatively, could analyze something like fossilized vs modern Alligator teeth to directly test this idea.

- Line 316: this difference has been highlighted by terminological differences for some time now in the literature (e.g. ziphodont vs. incrassate). The authors may prefer to just use the catch-all here, which I am also broadly supportive of, but I think they should at least acknowledge this terminology distinction somewhere.

- Line 421: missing an opening parenthesis here

Reviewer #3 (Remarks to the Author):

The paper of LeBlanc and coauthors describe a very meticulous characterization of chemical, structural and mechanical properties of reptile teeth. The key questions are yet evolutionary, namely to understand how iron-coated teeth evolved and adapted in reptiles.

I believe that the manuscript and the analytical work behind are overall well-done and definitely of interest. I do have some concerns about the LA-ICPMS analyses and the possible iron-coating for tyrannosaurid in figure 3 (or 2, based on the wrong numbering).

Here my specific comments:

- Figures: please check figure numbering, you currently have two figure(s) 1!

- References 22 and 12 are actually the same paper.

- Line 271: Zn role in regulating amelogenesis is known for e.g. humans (and mammals). Many works on high-spatial chemical resolution analyses of mammals' teeth show how outer enamel retains an enriched Zn-layer. I suggest the author to carefully read Müller et al. (2019, *Geochim Cosmochim Acta* 15:105-126), where Zn role is interpreted and described for humans. The slightly different distribution observed by LeBlanc et al. for Zn and Fe (Zn layer below Fe-pigmented layer) indicates likely different roles of the two elements in the metallomics of tooth enamel, maybe also linked to different stages of amelogenesis.

- Line 495: you need to report the measured isotopes (m/z) rather than the elements. Was the CRC used for all the analytes or Fe only? Please specify.

- Line 510: I'm not convinced by this approach. I think that overall is ok, but I find much more robust the protocol commonly employed in the geochemical community, namely reporting cps as normalized intensities to Ca. This means that you should reports maps as Sr/Ca, Fe/Ca, Zn/Ca and so on. At least, you should also report element/Ca maps and compare them with your corrected-maps to see if

7they are similar.

- Diagenesis. I understand that you would expect the enamel of tyrannosaurids to have a similar elemental distribution to that of other reptiles (and mammals), i.e., a Zn-Fe rich outer layer; I would think so too. However, if you cannot rule out that the Fe and Zn distributions in Figure 3 are not the result of a diagenetic alteration of the outer enamel, it is difficult to fully support the hypothesis. Some diagenetic markers (such as Y among those measured) or easily altered elements such as Ba should be reported in the same way. If these show a different distribution, then your statement will be more solid. For example, in suppl Figure 9, Fe (c) and Y (f) show a similar distribution, likely resulting from diagenetic uptake.

Author Rebuttal to Initial comments

Reviewer #1

The paper of LeBlanc et al. documents in detail the dental enamel in one of the most iconic extant reptiles, *Varanus komodoensis*. Using different, cutting-edge, technologies, the authors found that the teeth of this lizard possess orange, iron- and zinc-enriched coatings on their tooth serrations and tips. The authors further compare and discuss the potential of this case in other extant but also extinct reptiles (as for the latter though, strangely they confine their investigations to archosaurs, with no mention of the possibility of that case in extinct anguimorph [especially varanoid] lizards [see my comments below]). The manuscript is overall well written and the advanced technological methods presented therein seem to justify the conclusions of the authors. Some general comments:

The authors were probably unaware of the very recently published paper of Georgalis et al. (2023), where there was documentation of the pulp cavity of extant *Varanus* and *Lanthanotus* with the aid of μ CT scanning, as well as a detailed survey of the tooth serration in varanids, covering a wide range of extant taxa. I therefore consider that this publication should be mentioned.

Georgalis, G.L., B. Mennecart and K.T. Smith. 2023. First fossil record of *Varanus* (Reptilia, Squamata) from Switzerland and the earliest occurrences of the genus in Europe. *Swiss Journal of Geosciences* 116:9 (10 pp.).

Response:

We have included this in an updated comparison with other varanids. Also note that we have added Figure 3- a comparison of enamel pigmentation across several *Varanus* species alongside images of the serrations in fossil *Varanus priscus* ("Megalania") teeth to demonstrate that fossilization masks our ability to visually detect iron coatings in varanids. We would expect it to be there, given that it is most prominent in large varanids with serrated teeth, however we found no evidence for this after visual inspection of specimens at the South Australia Museum (SAM) by A. LeBlanc in the summer of 2023.

Also, I suggest mentioning somewhere (perhaps in the Introduction or the Discussion) a couple of sentences about the presence of serrated teeth in fossil varanids. This has been investigated in detail in Georgalis et al. (2019) for European fossil varanids, where all such occurrences of serrated teeth from the continent were documented (a further occurrence with serrated teeth was subsequently described in the above mentioned Georgalis et al. 2023 paper). Moreover, in the same paper of Georgalis et al. (2019), there were fossil varanid teeth from the same Neogene locality in Greece that had four different kinds of tooth serration: teeth either completely serrated (both mesially and distally), completely non-serrated, serrated mesially but not distally, and serrated distally but not mesially – although the authors stated that it was unclear whether these differences constituted some kind of intraspecific variation or diagnostic features or even taphonomic alteration, I consider that a sentence about this Greek occurrence should be mentioned in the current manuscript.

Georgalis, G.L., A. Villa, M. Ivanov, D. Vasilyan and M. Delfino. 2019. Fossil amphibians and reptiles from the Neogene locality of Maramena (Greece), the most diverse European herpetofauna at the Miocene/Pliocene transition boundary. *Palaeontologia Electronica* 22.3.68:1–99. <https://palaeo-electronica.org/content/2019/2797-fossil-herpetofauna-maramena>

Response:

We have added several of the reviewer's suggested references regarding the fossil record of varanid teeth with serrations and include a statement on the prevalence of pigmented

cutting edges in an updated sample of *Varanus* teeth, which includes additional taxa studied by LeBlanc at the SAM. Please note that we are extremely limited in terms of manuscript length, and so we have summarized these occurrences briefly. We do however illustrate the different arrangements of cutting edges and pigmentations among extant *Varanus* and the effects of fossilization on the colouration of varanid teeth using the new Figure 3 (while also citing several of the suggested references). There are closeup images of most of the extant varanid teeth we studied in the Supplementary Information. The updated main text now reads:

*“Given the surprising and consistent orange colouration of both the replacement and functional teeth in *V. komodoensis*, we further surveyed skeletal collections to determine the prevalence of this feature across a sample of *Varanus* species (Fig. 3a). Orange cutting edges were present in *V. salvadorii*, *V. rosenbergi*, and *V. giganteus* to varying degrees, and occasionally in *V. varius*, *V. salvator* and *V. indicus* teeth (Supplementary Fig. 4). These species all bear serrations, albeit smaller than those of *V. komodoensis*. Pigmented cutting edges were not visible in five other *Varanus* species, including some with small serrations and others that had non-serrated cutting edges. For comparisons, the cutting edges of the teeth of the anguimorph lizard *Heloderma* and a small sample of other squamate specimens we examined were not pigmented (Supplementary Fig. 5; Extended Data 1). This suggests that iron-pigmented cutting edges are found in several ziphodont *Varanus* species, but are most pronounced in *V. komodoensis*.*

*The prevalence of pigmented cutting edges, particularly in the larger serrated varanid teeth, led us to further predict that the largest ziphodont squamate, the extinct *Varanus priscus* (“Megalania”) may have also possessed iron-coated serrations. However, these fossilized teeth showed no visual evidence of pigmentation (Fig. 3b, c). Indeed, the varanid fossil record in general includes many fossilized teeth bearing serrations^{17–21}, however none of these occurrences show any obvious pigmentation similar to their extant counterparts. Furthermore, the small absolute sizes of varanid teeth and the paucity of fossil teeth available for destructive analyses make it difficult to assess the material properties of iron coatings in extant taxa, or how fossilization has hindered our ability to visually detect them in extinct species. We therefore sought to test for iron-pigmented enamel in another reptilian group with much larger teeth and a more abundant dental fossil record: modern and fossil crocodylians and the ziphodont teeth of theropod dinosaurs”*

We believe this re-written section also better transitions the manuscript into our investigation of the effects of fossilization on the chemistry of reptilian enamels in crocodylians, where we had access to far more extant and fossil samples for destructive analyses.

In the section “Pigmented cutting edges in other extant reptiles” (or at least in the Discussion), I would suggest mentioning something also (what is known about the cutting edges) for *Lanthanotus* but also for *Shinisaurus* (perhaps McDowell and Bogert 1954). Also, for helodermatids, I suggest citing Bhullar and Smith (2008). Most importantly though, here, the authors “jump” directly from the extant lizards to the Cretaceous dinosaurs. It would be so good, in the intermediate, to mention what is the case with cutting edges in extinct varanoids (or even anguimorphs in general). I would definitely cite about the European Paleogene form *Palaeovaranus* (see Georgalis and Scheyer 2019; Georgalis et al. 2021). What about the situation in the North American and European *Saniwa* (see Smith 2006; Augé et al. 2022) or the European *Paranecrosaurus* (see Smith and Habersetzer 2021) and *Melanosauroides* (see Georgalis 2017)? The authors should cite these respective primary figures for these taxa and state the shape of their cutting edges. Of course, the authors claim that their primary goal is to check these results in theropods, however, I still consider that, even this basic mention of extinct squamates need to be done, especially when the primary study object of the manuscript is a squamate. I therefore strongly suggest that a couple of sentences should be inserted here about these fossil varanoid taxa, with the appropriate relevant references I indicated (so that the reader can directly compare the figures in the original papers).

Augé, M.L., A. Folie, R. Smith, A. Phélizon, P. Gigase and T. Smith. 2022. Revision of the oldest varanid, *Saniwa orsmaelensis* Dollo, 1923, from the earliest Eocene of northwest Europe. *Comptes Rendus Palevol* 21(25):511–529.

Bhullar, B-A.S. and Smith, K. T. 2008. Helodermatid lizard for the Miocene of Florida, the evolution of the dentary in Helodermatidae, and comments on dentary morphology in Varanoidea. *Journal of Herpetology* 42:286–302.

Georgalis, G.L. 2017. *Necrosaurus* or *Palaeovaranus*? Appropriate nomenclature and taxonomic content of an enigmatic fossil lizard clade (Squamata). *Annales de Paléontologie* 103:293–303.

Georgalis, G.L., A. Čerňanský and J. Klembara. 2021. Osteological atlas of new lizards from the Phosphorites du Quercy (France), based on historical, forgotten, fossil material. *Geodiversitas* 43(9):219–293.

Georgalis, G.L. and T.M. Scheyer. 2019. A new species of Palaeopython (Serpentes) and other extinct squamates from the Eocene of Dielsdorf (Zurich, Switzerland). *Swiss Journal of Geosciences* 112:383–417.

McDowell S. B. and Bogert C. M. 1954. The systematic position of Lanthanotus and the affinities of the anguinomorphans lizards. *Bulletin of the American Museum of Natural History* 105(1):1–142.

Smith K.T. 2006. A diverse new assemblage of late Eocene squamates (Reptilia) from the Chadron formation of North Dakota, USA. *Palaeontologia Electronica* 9(2):5A:1–44.
Smith K.T. and J. Habersetzer. 2021. The anatomy, phylogenetic relationships, and autecology of the carnivorous lizard “Saniwa” feisti Stritzke, 1983 from the Eocene of Messel, Germany. *Comptes Rendus Palevol* 20(23):441–506.

Response:

Again, given the length requirements for Nature Ecology and Evolution (we are at the word limit now) and the need to address several key points raised by the other reviewers, we are unable to explore this beyond our study of *Varanus* and a very small sample of other squamates (which was initially done only to determine if orange-coloured enamel was universally present across squamates regardless of tooth shape). Our initial focus with this study, as pointed out by the reviewer, was the structure and chemistry of the enamel in ziphodont reptiles, with the Komodo dragon acting as the canonical modern analogue. This will undoubtedly lead to future investigations into the enamel chemistry and structure of other extant reptiles, but for this study our focus is on this initial discovery and its implications for the hidden complexity of carnivorous reptile enamel. We do look forward to exploring this in other species in the future!

Finally, what about this “Agamidae indet.” and “Iguanidae indet.” from the supplementary

material? There is no further information, geographic provenance, age (fossil or extant)? If extant, why the authors did not use some precisely identified (to the species level) specimen?

Response:

These specimens are derived from a small museum collection at King's College London's Museum of Life Sciences. This makes them amenable to high-resolution imaging using our in-house facilities, but unfortunately, the taxonomic identity of these specimens was not recorded in the collections database. We can only provide the limited information we have from the museum. However, the message remains the same: outside of varanids, the presence of pigmented enamel is unexplored, but there are no immediately obvious examples that we could find and squamate teeth are not normally pigmented orange.

I consider that the authors will be able to address these important issues I raised above. If so, following this moderate revision, I will be happy to recommend this manuscript for publication. I am at your disposal for any further query. Yours sincerely Dr. Georgios Georgalis

Reviewer #2

I think this paper by LeBlanc et al. is very interesting, as are the results. It represents a detailed application of well-established methods to provide a refreshing viewpoint and test of some historical questions/hypotheses concerning tooth structure and use in reptiles. The results are detailed and often backed up by multiple lines of analytical evidence (which combine to be a truly impressive amount of supplementary data). Nonetheless, there are some areas where it can be improved, or where additional comparisons would strengthen the results, particularly as they pertain to the Discussion, which I feel is perhaps the overall weakest section.

I have several minor comments, and then two more serious concerns/comments. One relates to the structure of the figures and their relation to the flow/readability of the paper, and the other relates to a lack of discussion of several topics that have bearing on the strength of the conclusions.

Provided these issues are addressed, I would recommend the paper be accepted.

Structure + Figures

These next points may be primarily an imposition of the sort of limits on figure number and style by the journal, but either way, I found the figures in this paper cluttered and at times frustrating to read, since they are very dense with information and are often lettered in a way that does not flow in any consistent manner (i.e. up-down, left-right, clockwise-anticlockwise, etc). They can of course be understood with some effort, but I think the text + figures + data are structured together in a way that almost actively works against the reader following the authors' arguments without having to constantly check back and re-read sections to understand what point they are trying to get across. If the information was presented in a way that was more accessible, then I think it would be much more straightforward to follow the arguments made in the text by the authors (and I'll note that I thought the text on its own was much better, so this issue is more with the figures themselves and the integration of figure content with text points).

If it is possible, I'd suggest some of the very dense figures be split into multiple figures so that the information can be presented to readers in a more step-wise fashion, rather than all at once via the current results plates. Alternatively, one could at least add in a figure or two to synthesize the information from the plates and multiple analytical lines into summary diagrams, with readers directed to the supplement to see the various more specific outputs across the various image plates (potentially moving some of the current plates to supplement to make room if figure count is too limiting).

I understand that there is a lot going on in the paper and as a result there are many moving parts that all have to be reported and displayed in some way, so if it ends up that the changes I suggested above for figures cannot be implemented, so be it. I just wanted to mention it because I found it did impact my reading and increased the time needed to parse through what is otherwise a very solid study.

Response:

We were able to expand the number of in-text figures from 4 to 6 and in the process have made great effort to simplify and reduce the clutter within the existing figures. The following changes have been made:

Fig. 2: We have removed the zinc and composite elemental maps from the figure and restrict the figure to iron and calcium mapping, which is the main point of the study anyways. We have also created more consistent colouring between elemental maps generated from XRF and SEM analyses (i.e., orange for iron and green/teal for calcium). We have also generated new and clearer TEM images of the enamel-iron coating interface (Fig. 2I-n).

Fig. 3: This new figure illustrates the diversity of serration and tooth morphologies across several species of *Varanus*, illustrating the distribution of enamel pigmentation across the genus. This figure also illustrates why we would expect to find iron-pigmented serrations in the giant fossil varanid *Varanus priscus* ("Megalania"). This serves as the transition point for our following figures/results focused on the confounding effects of fossilization.

Fig. 4: We have now split the crocodylian and theropod elemental mapping into two separate figures. Fig. 4 now illustrates the distribution of iron along the unserrated cutting edges in three species of extant crocodylian and includes new maps from a fossil crocodylian tooth (from Dinosaur Provincial Park, similar to the theropod teeth in subsequent figures) to illustrate the effects of fossilization on the distribution of iron and calcium in extant vs fossil crocodylian teeth. We use this as support for why we may not see pigmented cutting edges in our small

sample of fossil *Varanus priscus* teeth and why (later in the paper) we think we cannot find iron pigmentation in dinosaur teeth even when we might initially expect to see it. We have also removed the zinc maps and concentrate solely on calcium and iron for consistency with Fig. 2.

Fig. 5: This figure presents only the elemental data for fossil theropods: tyrannosaurids and a new sample of fossil dromaeosaurid teeth- all from the Late Cretaceous of Dinosaur Provincial Park, Canada. The dromaeosaurid tooth samples were borrowed and analyzed in response to this reviewer's comments below.

Fig. 6: Panels have been removed and rearranged for improved clarity. Diffraction patterns are now only illustrating the differences between the wavy enamel within the tyrannosaurid tooth serrations, and the columnar enamel found elsewhere along the tooth crown. This is done to better illustrate the utility of diffraction-based mapping of reptile enamel: it can quantify variation in prismless enamel microstructures along a reptile tooth crown. We have also increased the sizes of the enamel diffraction maps for improved readability. The lettering here has also been changed to give it more logical ordering (we agree, this was confusing in the first version). We hope this figure is clearer now.

Results & Interpretations

I have some concerns with the interpretations of the results and integration with the Discussion, at least insofar as they appear to ignore several sources of variation/uncertainty.

Many of these potential sources of uncertainty are acknowledged by the authors in the text, so I don't think they go undiscussed because the authors had not considered them. But I think they do need to be addressed more directly, whether through more extensive discussions and literature comparisons, or by some additional analyses, lest aspects of the otherwise very interesting conclusions of this paper be unfairly framed as a 'just-so' story. I'm sure the authors have thought about these topics, so I imagine that their overall

16conclusions will not be strongly impacted by including discussion of them, and their inclusion should only strengthen the overall impact of the paper.

These sources of uncertainty and broader discussion include the following:

l) The lack of comparison to theropods that are of similar size to *Varanus*. In the Discussion, you argue that “Alternative feeding strategies between *V. komodoensis* (de-fleshing food items) and some tyrannosaurids (scoring and ingesting bone as well as the flesh) may explain these divergent strategies for maintaining serrated teeth, but these differences could also reflect scaling effects between these two ziphodonts. Despite both having relatively thin enamel coatings, only the large teeth of theropod dinosaurs possess enough enamel over which the microstructure can vary; the enamel is too thin in extant ziphodont lizards to permit structural variations on a similar scale”, but at no point do you provide comparisons to smaller theropods (such as dromaeosaurs), which would seem to be the perfect test of the two hypotheses offered above by the authors, while avoiding the various size-based uncertainties introduced by limiting comparisons to larger theropods such as tyrannosaurs.

Response:

We agreed with this suggestion and have now collected data to address this. We have loaned several dromaeosaurid teeth from the University of Alberta’s collection of Dinosaur Provincial Park theropod teeth and have subjected them to the same offline (non-synchrotron, due to time constraints) elemental mapping as well as SEM imaging as was done for the tyrannosaurid teeth.

The results provided some support for an influence of absolute tooth size on the presence or absence of complex serration enamel (although, ruling out phylogenetic differences is still not possible, given the limited taxonomic and ontogenetic variation covered by this sample, but this would vastly increase the length and breadth of this manuscript beyond what was originally intended).

Our sample of small (~*Varanus*-sized) theropod teeth from Dinosaur Provincial Park do not show the same wavy enamel under SEM as the teeth of the larger tyrannosaurid dinosaur teeth. However, we did observe some qualitative differences in one tooth from the serration and non-serration enamel (Supplementary Fig. 16k–m). Coincidentally, this particular tooth had the thickest enamel of any of our dromaeosaurid sample.

We now raise this in the discussion in support of our hypotheses regarding the evolution of serration enamel complexity:

*“Alternative feeding strategies between *V. komodoensis*⁹ and some tyrannosaurids^{10,36} may explain these divergent strategies for maintaining serrated teeth, but these differences could also reflect scaling effects between these two ziphodont taxa. Dromaeosaurid dinosaur teeth provide partial evidence for this. The dromaeosaurid teeth sampled in this study were of similar sizes to those of *V. komodoensis* and most showed comparably thin and simple enamel microstructure between the serrations and the rest of the crown (Supplementary Fig. 16). Only in a dromaeosaurid tooth with thicker enamel (40 μm or more) did we see any qualitative differences in enamel structure on- and off-serration, but these were still not as pronounced as in the larger tyrannosaurid teeth (Supplementary Fig. 16k–o). Despite having relatively thin enamel coatings, it is possible that only the larger teeth of theropod dinosaurs possess enough enamel over which the microstructure can influence the mechanical wear of the serrations. Conversely, the enamel is likely too thin in extant ziphodont lizards to influence tooth wear on a similar scale to tyrannosaurids. The thin, structurally simple enamel in *V. komodoensis* probably plays a limited role in the cutting function of each tooth, and natural selection may have alternatively favoured individuals with more elaborate iron coatings to mitigate wear along their rapidly replaced teeth. If theropod dinosaurs did similarly incorporate iron into their ziphodont teeth, then we may expect to find clearer evidence in smaller species or individuals with comparably thin enamel. Identifying these iron coatings in fossil reptile teeth, however, remains a challenge; we did not see any evidence for iron sequestration, even in small dromaeosaurid teeth.”*

II) The lack of analytical comparisons to other squamates, and particularly a lack of discussion of the implications of Fe coatings being only variably present in different species of *Varanus*. To better support the interpretation of these coatings as an adaptation for reinforcing cutting teeth in komodo dragons, and contrasting them against what is being done in different archosaurs, I think it is reasonably important to explain why this feature may be so variable within *Varanus* itself, since that would presumably impact the enamel

properties of these other *Varanus* species outside of *komodoensis*. You do note that it may just be that at small size the visual effect isn't noticeable, but again, some direct analytical data on this would go a long way to supporting the core argument of the conclusions.

Response:

This is an excellent question. We have collected additional data on the visual occurrences of orange pigmentation along the serrations of large Australian monitor lizards during a collections visit to the South Australian Museum by LeBlanc in 2023. These data are now also presented in a summary cladogram in the revised Fig. 3 to illustrate the distribution of serration pigmentation across our *Varanus* sample. Here we now illustrate which species seem to consistently show orange cutting edges, which seem to only occasionally show it, and which never seem to show it. We are also more explicit in the discussion when we discuss and interpret this difference between varanids and crocodylians:

*"Our ability to visually detect the same pigmented cutting edges in closely related species of *Varanus* suggests that iron sequestration may also be widespread in reptile teeth. We therefore predicted to see this feature in the large ziphodont teeth of closely related extinct taxa as well [e.g., *Varanus* ("Megalania") *priscus*]. However, our broader comparisons show that iron sequestration is neither a presence-or-absence phenomenon in extant reptiles, nor is it easily detectable in fossil reptile teeth. Even where pigmentation was absent from the cutting edges in extant reptile teeth under white light, fluorescence imaging techniques revealed that even extant crocodylians can possess iron-enriched outer enamel layers, particularly along the cutting edges. This suggests many reptiles may have iron-enriched enamel, but that only some species have evolved prominent iron coatings along specific parts of their tooth crowns, presumably as feeding adaptations. Under high enough concentrations, the iron layers are visible under white light, as is the case in *V. komodoensis* and some of its ziphodont relatives."*

We have also updated Extended Data 1 and made it clearer in which specimens/taxa we see this pigmentation, but we stress in the discussion that this is not a presence-absence phenomenon. It is likely influenced by size of the tooth and concentration along the serrations/carinae, and maybe even ontogeny (though this is much harder to demonstrate without more synchrotron and LA-ICP-MS analyses of juvenile Komodo dragons- which requires destructive sampling). It can also be invisible in

white light (as we report in crocodylians), but still be consistently present using LSF or XRF imaging. Komodo dragons are by far the most consistent and most conspicuous, but only with detailed, destructive sampling, could this particular concern be addressed.

However, diving into the quantitative variations in iron concentrations within the outer 500-1000 nanometers of different *Varanus* species would take this study into a completely different trajectory than was originally intended and be prohibitively expensive (FIB milling, LA-ICP-MS time, etc.). While this would help address this particular query, this is also well beyond what we are able to achieve in any reasonable timeline (this would require years of collections loans, synchrotron time, and data analysis). Our intention here is to report these unusual phenomena and follow this with more detailed research endeavours in the future, like this query the Reviewer mentions.

III) The lack of direct mammalian comparisons (or other way to assess impact of tooth replacement as a component of the strategy underscoring the management of tooth wear). To be clear, their existence is mentioned, and some minor differences are noted, but other than that it is barely examined. This seems unusual to me given how it would represent a fairly direct analogue of this sort of adaptation in a distant group of amniotes. Along the same lines, tooth replacement rates and strategies of polyphyodonty vs. diphyodonty, and the role they may play in questions of tooth function, tooth wear rates, and enamel thicknesses, etc. are also almost mostly undiscussed, even though how frequently the teeth are being replaced would presumably have a factor on selective pressures if the argument is that Fe concentration in enamel is an adaptation to strengthen cutting surfaces and reduce tooth wear in taxa with thin enamel.

Response:

We have attempted to address this in three ways:

- (1) We have conducted an additional synchrotron experiment using Iron-X-Ray Absorption Near-Edge Structure (Fe-XANES) to compare the iron layers in a beaver (*Castor canadensis*), the Komodo dragon, and a crocodylian (See Supplementary Fig. 9). With this we determined that Komodo dragons sequester iron as a coating of ferrihydrite, but even this appears to differ relative to the iron layers in beavers and crocodylians, suggesting alternative approaches to iron-enrichment of enamel across amniotes. Finding an analogue for the iron coatings in Komodo dragons, may be difficult, because it now appears there are multiple types of iron sequestration across amniotes.

Nevertheless, we now report in the results that the iron is sequestered as a ferrihydrite coating, and in the discussion, we now write:

“Furthermore, unlike in pigmented rodent teeth where mixed-phase iron oxides are incorporated into the intergranular spaces within enamel, iron appears to be concentrated into a distinct coating of ferrihydrite that is bonded to the underlying crystalline enamel in V. komodoensis (Fig. 2; Supplementary Fig. 9). This was unexpected, given that V. komodoensis possesses only 15-20 μm of enamel and replaces its teeth rapidly¹².”

- (2) We conducted another STEM and STEM-EDS session on a FIB-milled sample of the komodo dragon enamel. We present these in a revised Fig. 2 to more clearly illustrate the interface between the enamel and the iron coating (Fig. 2n) as well as the enamel-free coating that is rich in iron (Fig. 2l, m). This coating differs from the intragranular sequestration of iron oxides *within* beaver enamel- in Komodo dragons, they somehow develop an enamel-free coating of ferrihydrite. We describe this in the revised results.

- (3) Moving into a discussion of the effects of diphyodonty vs polyphyodonty on the prevalence of these features is a bit premature at this stage, but we attempted to address some facets of this in a revised part of the discussion.

At present, this paper is describing iron sequestration on the teeth of a lizard that apparently has one of the most rapid rates of tooth replacement within

squamates (Maho and Reisz, 2024: “Exceptionally rapid tooth development and ontogenetic changes in the feeding apparatus of the Komodo dragon”). In other words, the relationship between tooth replacement frequency and the presence or absence of iron is not clear, because a mammal does not replace its iron-coated adult tooth, whereas a Komodo dragon may replace it in several weeks (which makes this an exciting future direction of enquiry).

We now note the following in the revised discussion:

Given the new data on dromaeosaurid enamel we have collected, we have also updated the discussion on the effects of absolute tooth size and enamel thickness on the type of adaptations we have observed in ziphodont reptile enamel. This is quoted in response to this Reviewer’s point II) above. These factors may be more closely related to the appearance of iron-coated teeth in varanids than the replacement rate.

*“Despite having relatively thin enamel coatings, it is possible that only the larger teeth of theropod dinosaurs possess enough enamel over which the microstructure can influence the mechanical wear of the serrations. Conversely, the enamel is likely too thin in extant ziphodont lizards to influence tooth wear on a similar scale to tyrannosaurids. The thin, structurally simple enamel in *V. komodoensis* probably plays a limited role in the cutting function of each tooth, and natural selection may have alternatively favoured individuals with more elaborate iron coatings to mitigate wear along their rapidly replaced teeth.”*

IV) A lack of comparison of fossil vs modern teeth of the same taxon (or close relative) to test some of the interpretations made concerning elemental concentration patterns in theropod tooth tissues being the result of fossilization. This one is less critical, and I don’t think it’s a bad conclusion at present, but I think it is something that could warrant an actual test, or at least some additional reference to literature data.

Response:

Unfortunately, we requested but did not have permission to loan and section fossil varanid teeth and we have already cited the only elemental studies of fossil vs modern

reptile teeth we could find. The best test for this would be the fossil *V. komodoensis* or *V. priscus* teeth from Australia, but these are rare and precious samples. We have now at least included images of *V. priscus* to illustrate the effect fossilization has on the appearance of the serrations in Figure 3.

However, we can test this preservational hypothesis in extant vs fossil crocodylians and this is what we have added to the revised manuscript, both in the main text and supplementary information. We have added additional data from a Synchrotron-X-Ray MicroFluorescence (S- μ XRF) experiment where we compared our extant crocodylian teeth to those of similar-shaped fossil crocodylian teeth from Dinosaur Provincial Park (Canada)- the same region from which the theropod teeth were sourced. These show that even when we expect to find iron-enriched outer enamel layers (as we did for all of our extant crocodylian teeth), we do not see any obvious signs of iron-enriched outer enamel in fossil samples from Dinosaur Provincial Park. This supports our hypothesis that even when we expect to see these outer iron-enriched enamel layers, diagenesis appears to obscure their signal. You will find these data in Fig. 4 and Suppl. Fig. 7.

Minor Comments

- There may be a problem with figure numbering in the text, as I noticed that figure 3 was being called in-text a few times where I'm fairly sure Figure 2 was what was intended (for example, on Line 170). Similarly, Extended Data 2 is cited in text, but as far as I can tell it is not one of the files uploaded with the paper? Unless it is also synonymous with the Supplementary Information file (though at least some of what is mentioned in text for Extended Data 2 does not appear to be present in the Supp Info file). Perhaps everything is present but the order of things was changed at some point and not all of the old refs were caught? In any case, I'd recommend the authors go through and confirm all of their figures, supp figures, and extended data are present and being correctly cross-referenced in the text just in case this also happened elsewhere.

Response:

23We hope this has been corrected over the course of editing and uploading this version of the manuscript. Thank you for raising this issue. Extended Data 2 is a large table summarizing each specimen used for each synchrotron experiment, and the figure number(s) where they are used.

- Lines 258-260: this is probably better in the Discussion rather than Results, since it is mostly a speculation rather than something directly based on data. Alternatively, could analyze something like fossilized vs modern Alligator teeth to directly test this idea.

We have deleted the sentence.

- Line 316: this difference has been highlighted by terminological differences for some time now in the literature (e.g. zipodont vs. incrassate). The authors may prefer to just use the catch-all here, which I am also broadly supportive of, but I think they should at least acknowledge this terminology distinction somewhere.

Response:

The term “zipodonts” has been removed whenever referring to a zipodont taxon. However, as far as the authors can tell, “incrassate” seems specific to the rounded teeth of larger tyrannosaurid dinosaurs? Nearly all of the theropod teeth sectioned and subjected to elemental/diffraction analyses in this study were laterally compressed, blade-shaped teeth, similar to *V. komodoensis*. We feel introducing the term “incrassate” may confuse a more general readership in this case.

- Line 421: missing an opening parenthesis here

Corrected.

Reviewer #3

The paper of LeBlanc and coauthors describe a very meticulous characterization of chemical, structural and mechanical properties of reptile teeth. The key questions are yet evolutionary, namely to understand how iron-coated teeth evolved and adapted in reptiles. I believe that the manuscript and the analytical work behind are overall well-done and definitely of interest. I do have some concerns about the LA-ICPMS analyses and the possible iron-coating for tyrannosaurid in figure 3 (or 2, based on the wrong numbering).

Here my specific comments:

- Figures: please check figure numbering, you currently have two figure(s) 1!

Response:

We hope this has been corrected through the major revisions to the main text.

-References 22 and 12 are actually the same paper.

Response:

This has been corrected.

- Line 271: Zn role in regulating amelogenesis is known for e.g. humans (and mammals). Many works on high-spatial chemical resolution analyses of mammals' teeth show how outer enamel retains an enriched Zn-layer. I suggest the author to carefully read Müller et al. (2019, *Geochim Cosmochim Acta* 15:105-126), where Zn role is interpreted and described for humans. The slightly different distribution observed by LeBlanc et al. for Zn and Fe (Zn layer below Fe-pigmented layer) indicates likely different roles of the two elements in the metallomics of tooth enamel, maybe also linked to different stages of amelogenesis.

Response:

Many thanks for bringing these studies to our attention. The zinc story has been downplayed in the main text of this version of the manuscript, mainly because we cannot say for certain at this stage of our research programme exactly where zinc is located within the enamel/iron coating of Komodo dragons. We know it is consistently collocated with the iron-enriched regions of the teeth in our XRF and LA-ICP-MS maps, but after repeated attempts, we cannot isolate where within the thin veneer of enamel the zinc is located relative to the iron coating under SEM or TEM-EDS. For the sake of the length requirements and the need for future investigations into the development of the enamel and iron layer in reptiles (as noted by this reviewer), we only include the zinc maps in the supplementary information and make short references to the zinc distributions in modern reptile enamel. This includes acknowledging that the zinc may be a product of enamel secretion and maturation in the Results, following Muller et al. (2019).

- Line 495: you need to report the measured isotopes (m/z) rather than the elements. Was the CRC used for all the analytes or Fe only? Please specify.

Response: Again, thanks to the reviewer for bringing this to our attention. We have now edited the Materials and Methods to clarify our methodology and rationale for the chosen isotopes:

“The selected isotopes of interest ^{24}Mg , $^{44}\text{Ca}^{16}\text{O}$, ^{56}Fe , ^{66}Zn , $^{88}\text{Sr}^{16}\text{O}$, $^{89}\text{Y}^{16}\text{O}$ and $^{138}\text{Ba}^{16}\text{O}$ were chosen to maximise sensitivity whilst minimising isobaric/polyatomic interferences and increasing signal-to-noise ratio. Dynamic Reaction Collision (DRC) mode with oxygen as the reaction gas was employed in the collision reaction cell (CRC) to reduce the contribution of polyatomic interferences on imaging for all isotopes. The reactive oxygen species measured are proxies for our elements of interest (Ca, Sr, Y, Ba) and are therefore labelled in the LA-ICP-MS maps accordingly. To correct for instrumental drift, a series of NIST 612 standard ablation scans were performed before and after experimental samples.”

- Line 510: I'm not convinced by this approach. I think that overall is ok, but I find much more robust the protocol commonly employed in the geochemical community, namely reporting cps as normalized intensities to Ca. This means that you should reports maps as

Sr/Ca, Fe/Ca, Zn/Ca and so on. At least, you should also report element/Ca maps and compare them with your corrected-maps to see if they are similar.

We encountered two problems when attempting to use this approach:

The first issue relates to the different hardnesses of the enamel vs dentine, which always skew the amount of material we ablate from each tissue in a modern reptile tooth (we did not notice this issue in our fossil samples). We calculated that we were ablating >5x more dentine than enamel in modern tooth samples (see Supplementary Information and Materials and Methods). This means that even if we normalize each pixel in a map by Ca, we are still introducing the same bias between the two tissues (dentine and enamel) into each image (see figure to the right). Ironically, it was only through the unusual distributions of Ca in the original uncorrected maps that we realized this technique provides biased counts for dentine vs enamel.

A Fe/Ca map generated for enamel and dentine in a crocodylian tooth sample. Dentine is towards the bottom. The iron layer is in the outer edge of the enamel towards the top of the image (based on other techniques). Normalizing by Ca over-emphasizes iron counts throughout the enamel due to lower overall counts of Ca in the ablated enamel compared to dentine, and creates artefacts in the iron layers where there was no measurable Ca.

We noticed a second issue after attempting to standardize to Ca following the reviewer's comment (see figure above). This relates to the fact that some of the iron layer is devoid of calcium- which creates a situation where the region of interest includes an Fe count divided by zero. The resulting artefact in the maps is associated with a nonsensical ratio in an iron-rich, calcium-free coating of the enamel. Unfortunately it is the natural heterogeneity of these samples that limits our ability to standardize counts by Ca.

After these considerations, we have decided to retain our original protocol in the generation of our previous and new LA-ICP-MS maps, which is recommended by

27others who have done elemental work on variably ablated samples (Mervic, K., Elteren, J. T., Bele, M., Sala, M. 2024. "Utilizing ablation volume for calibration in LA-ICP-MS mapping to address variations in ablation rates within and between maps. *Talanta* 269: 125379).

- Diagenesis. I understand that you would expect the enamel of tyrannosaurids to have a similar elemental distribution to that of other reptiles (and mammals), i.e., a Zn-Fe rich outer layer; I would think so too. However, if you cannot rule out that the Fe and Zn distributions in Figure 3 are not the result of a diagenetic alteration of the outer enamel, it is difficult to fully support the hypothesis. Some diagenetic markers (such as Y among those measured) or easily altered elements such as Ba should be reported in the same way. If these show a different distribution, then your statement will be more solid. For example, in suppl Figure 9, Fe (c) and Y (f) show a similar distribution, likely resulting from diagenetic uptake.

Response:

We agree with the reviewer. Iron and zinc distributions can appear tantalizingly similar in some of the maps we generated for tyrannosaurid (and now dromaeosaurid-see Supplementary Information) serration enamel, but these are not consistent across different planes of section. The maps generated through horizontal sections do not show a sequestration of iron within the enamel in any way similar to *V. komodoensis* (original data) or extant crocodylians (new data in revised figures). We have modified the text accordingly and point out that at present, we cannot see or find evidence for iron coatings in fossil varanid, crocodylian, or theropod teeth. This does not mean that it was not there in life, but that diagenesis would have obscured this information and we must seek other correlates for this in future studies.

The text in the Results and Discussion, as well as the original figure in question, have all been modified to address this.

Decision Letter, first revision:

2820th March 2024

Dear Dr. LeBlanc,

Thank you for submitting your revised manuscript "Iron-coated Komodo dragon teeth and the complex enamel of carnivorous reptiles" (NATECOLEVOL-23071631A). It has now been seen again by the original reviewers and their comments are below. The reviewers find that the paper has improved in revision, and therefore we'll be happy in principle to publish it in Nature Ecology & Evolution, pending minor revisions to satisfy the reviewers' final requests and to comply with our editorial and formatting guidelines.

[REDACTED]

Reviewer #1 (Remarks to the Author):

Dear Editor, dear Authors,

I read the revised version of the manuscript and I am glad to see that the Authors undertook many of my previous suggestions. I realize that some of my other suggestions I indicated in the previous round were not followed due to word limits within the manuscript. This is unfortunate, especially the comparisons with extinct close relatives of varanids, however, I understand that.

I have now only to comment about some misspellings of the names *Crocodylus porosus* on "21243_1_data_set_228796_s9h1yq" and *Megalosaurus bucklandii* and *Osteolaemus tetraspis* on "21243_1_supp_229292_s9q2yg"

I therefore recommend the paper for publication.

Yours sincerely

Dr. Georgios Georgalis

Reviewer #2 (Remarks to the Author):

I've now completed my review of the revised version of this submission. I want to thank the authors for the thorough work they have done in addressing my comments. I am satisfied with the responses and new analyses, revised figures, and text changes. In the small number of places where they did not directly address an issue, I am satisfied with the reasons given for not doing so.

29Lastly, as a very minor point, regarding the comment I had made regarding ziphodont vs incrassate, I would agree with the authors that almost all the teeth sectioned would indeed be best described as ziphodont, and was mostly meaning that since tyrannosaur tooth shape has been considered sufficiently different to warrant another term in some studies, it might just be worth saying that once before using the term for all of them. Otherwise had no issue with the use of ziphodont as a descriptor for the majority of the teeth used in the study.

Overall, I think this represents a very interesting and thorough contribution, and would recommend it now be accepted for publication.

Reviewer #3 (Remarks to the Author):

I rechecked the whole manuscript and the author answers to my comments. I'm satisfied with the review and the overall job done by the authors. I think the MS is ready for publication.

Our ref: NATECOLEVOL-23071631A

10th April 2024

Dear Dr. LeBlanc,

Thank you for your patience as we've prepared the guidelines for final submission of your Nature Ecology & Evolution manuscript, "Iron-coated Komodo dragon teeth and the complex enamel of carnivorous reptiles" (NATECOLEVOL-23071631A). Please carefully follow the step-by-step instructions provided in the attached file, and add a response in each row of the table to indicate the changes that you have made. Please also check and comment on any additional marked-up edits we have proposed within the text. Ensuring that each point is addressed will help to ensure that your revised manuscript can be swiftly handed over to our production team.

****We would like to start working on your revised paper, with all of the requested files and forms, as soon as possible (preferably within two weeks). Please get in contact with us immediately if you anticipate it taking more than two weeks to submit these revised files.****

30When you upload your final materials, please include a point-by-point response to any remaining reviewer comments.

In recognition of the time and expertise our reviewers provide to Nature Ecology & Evolution's editorial process, we would like to formally acknowledge their contribution to the external peer review of your manuscript entitled "Iron-coated Komodo dragon teeth and the complex enamel of carnivorous reptiles". For those reviewers who give their assent, we will be publishing their names alongside the published article.

Nature Ecology & Evolution offers a Transparent Peer Review option for new original research manuscripts submitted after December 1st, 2019. As part of this initiative, we encourage our authors to support increased transparency into the peer review process by agreeing to have the reviewer comments, author rebuttal letters, and editorial decision letters published as a Supplementary item. When you submit your final files please clearly state in your cover letter whether or not you would like to participate in this initiative. Please note that failure to state your preference will result in delays in accepting your manuscript for publication.

Cover suggestions

We welcome submissions of artwork for consideration for our cover. For more information, please see our guide for cover artwork.

Nature Ecology & Evolution has now transitioned to a unified Rights Collection system which will allow our Author Services team to quickly and easily collect the rights and permissions required to publish your work. Approximately 10 days after your paper is formally accepted, you will receive an email in providing you with a link to complete the grant of rights. If your paper is eligible for Open Access, our Author Services team will also be in touch regarding any additional information that may be required to arrange payment for your article.

Please note that *Nature Ecology & Evolution* is a Transformative Journal (TJ). Authors may publish their research with us through the traditional subscription access route or make their paper immediately open access through payment of an article-processing charge (APC). Authors will not be

31required to make a final decision about access to their article until it has been accepted. Find out more about Transformative Journals

Authors may need to take specific actions to achieve compliance with funder and institutional open access mandates. If your research is supported by a funder that requires immediate open access (e.g. according to Plan S principles) then you should select the gold OA route, and we will direct you to the compliant route where possible. For authors selecting the subscription publication route, the journal's standard licensing terms will need to be accepted, including [a href="https://www.nature.com/nature-portfolio/editorial-policies/self-archiving-and-license-to-publish"](https://www.nature.com/nature-portfolio/editorial-policies/self-archiving-and-license-to-publish). Those licensing terms will supersede any other terms that the author or any third party may assert apply to any version of the manuscript.

[REDACTED]

[REDACTED]

Reviewer #1:

Remarks to the Author:

Dear Editor, dear Authors,

I read the revised version of the manuscript and I am glad to see that the Authors undertook many of my previous suggestions. I realize that some of my other suggestions I indicated in the previous round were not followed due to word limits within the manuscript. This is unfortunate, especially the comparisons with extinct close relatives of varanids, however, I understand that.

I have now only to comment about some misspellings of the names *Crocodylus porosus* on "21243_1_data_set_228796_s9h1yq" and *Megalosaurus bucklandii* and *Osteolaemus tetraspis* on "21243_1_supp_229292_s9q2yg"

I therefore recommend the paper for publication.

Yours sincerely

Dr. Georgios Georgalis

Reviewer #2:

Remarks to the Author:

32I've now completed my review of the revised version of this submission. I want to thank the authors for the thorough work they have done in addressing my comments. I am satisfied with the responses and new analyses, revised figures, and text changes. In the small number of places where they did not directly address an issue, I am satisfied with the reasons given for not doing so.

Lastly, as a very minor point, regarding the comment I had made regarding ziphodont vs incrassate, I would agree with the authors that almost all the teeth sectioned would indeed be best described as ziphodont, and was mostly meaning that since tyrannosaur tooth shape has been considered sufficiently different to warrant another term in some studies, it might just be worth saying that once before using the term for all of them. Otherwise had no issue with the use of ziphodont as a descriptor for the majority of the teeth used in the study.

Overall, I think this represents a very interesting and thorough contribution, and would recommend it now be accepted for publication.

Reviewer #3:

Remarks to the Author:

I rechecked the whole manuscript and the author answers to my comments.

I'm satisfied with the review and the overall job done by the authors.

I think the MS is ready for publication.

Author Rebuttal, first revision:

Reviewer #1

I read the revised version of the manuscript and I am glad to see that the Authors undertook many of my previous suggestions. I realize that some of my other suggestions I indicated in the previous round were not followed due to word limits within the manuscript. This is unfortunate, especially the comparisons with extinct close relatives of varanids, however, I understand that.

I have now only to comment about some misspellings of the names *Crocodylus porosus* on "21243_1_data_set_228796_s9h1yq" and *Megalosaurus bucklandii* and *Osteolaemus tetraspis* on "21243_1_supp_229292_s9q2yg" I therefore recommend the paper for publication.

Yours sincerely,

33Dr. Georgios Georgalis

Response: Many thanks for your understanding with this length limitation. We do appreciate this limitation to our comparative analyses and intend to follow through with more detailed investigations of extant and fossil squamate teeth in the near future. We believe we have caught all the above mentioned typos in the extended data and supplementary information as well.

Reviewer #2

I've now completed my review of the revised version of this submission. I want to thank the authors for the thorough work they have done in addressing my comments. I am satisfied with the responses and new analyses, revised figures, and text changes. In the small number of places where they did not directly address an issue, I am satisfied with the reasons given for not doing so.

Lastly, as a very minor point, regarding the comment I had made regarding zipodont vs incassate, I would agree with the authors that almost all the teeth sectioned would indeed be best described as zipodont, and was mostly meaning that since tyrannosaur tooth shape has been considered sufficiently different to warrant another term in some studies, it might just be worth saying that once before using the term for all of them. Otherwise had no issue with the use of zipodont as a descriptor for the majority of the teeth used in the study.

Overall, I think this represents a very interesting and thorough contribution, and would recommend it now be accepted for publication.

34Response: We appreciate the reviewer's careful critique of our study. It has led to significant improvements to the revised manuscript (as it has thanks to the other reviewers as well). For the sake of simplicity and the main take-home message regarding the nature of ziphodont teeth across reptiles, we have decided to retain the text as-is when describing this tooth type, now understanding that there exists different terminology for the more robust teeth of tyrannosaurids.

Reviewer #3

I rechecked the whole manuscript and the author answers to my comments. I'm satisfied with the review and the overall job done by the authors. I think the MS is ready for publication.

Response: Again, thank you for your constructive comments and suggestions in the previous round of review.

Final Decision Letter:

8th May 2024

Dear Professor Addison,

We are pleased to inform you that your Article entitled "Iron-coated Komodo dragon teeth and the complex enamel of carnivorous reptiles", has now been accepted for publication in Nature Ecology & Evolution.

Over the next few weeks, your paper will be copyedited to ensure that it conforms to Nature Ecology and Evolution style. Once your paper is typeset, you will receive an email with a link to choose the appropriate publishing options for your paper and our Author Services team will be in touch regarding any additional information that may be required

Due to the importance of these deadlines, we ask you please us know now whether you will be difficult to contact over the next month. If this is the case, we ask you provide us with the contact information (email, phone and fax) of someone who will be able to check the proofs on your behalf, and who will be available to address any last-minute problems . Once your paper has been scheduled for online

35publication, the Nature press office will be in touch to confirm the details.

Acceptance of your manuscript is conditional on all authors' agreement with our publication policies (see www.nature.com/authors/policies/index.html). In particular your manuscript must not be published elsewhere and there must be no announcement of the work to any media outlet until the publication date (the day on which it is uploaded onto our web site).

Please note that *Nature Ecology & Evolution* is a Transformative Journal (TJ). Authors may publish their research with us through the traditional subscription access route or make their paper immediately open access through payment of an article-processing charge (APC). Authors will not be required to make a final decision about access to their article until it has been accepted. Find out more about Transformative Journals

Authors may need to take specific actions to achieve compliance with funder and institutional open access mandates. If your research is supported by a funder that requires immediate open access (e.g. according to Plan S principles) then you should select the gold OA route, and we will direct you to the compliant route where possible. For authors selecting the subscription publication route, the journal's standard licensing terms will need to be accepted, including [a href="https://www.nature.com/nature-portfolio/editorial-policies/self-archiving-and-license-to-publish"](https://www.nature.com/nature-portfolio/editorial-policies/self-archiving-and-license-to-publish). Those licensing terms will supersede any other terms that the author or any third party may assert apply to any version of the manuscript.

We welcome the submission of potential cover material (including a short caption of around 40 words) related to your manuscript; suggestions should be sent to Nature Ecology & Evolution as electronic files (the image should be 300 dpi at 210 x 297 mm in either TIFF or JPEG format). Please note that such pictures should be selected more for their aesthetic appeal than for their scientific content, and that colour images work better than black and white or grayscale images. Please do not try to design a cover with the Nature Ecology & Evolution logo etc., and please do not submit composites of images related to your work. I am sure you will understand that we cannot make any promise as to whether any of your suggestions might be selected for the cover of the journal.

To assist our authors in disseminating their research to the broader community, our SharedIt initiative provides you with a unique shareable link that will allow anyone (with or without a subscription) to

read the published article. Recipients of the link with a subscription will also be able to download and print the PDF.

You can generate the link yourself when you receive your article DOI by entering it here: <http://authors.springernature.com/share>.

[REDACTED]

P.S. Click on the following link if you would like to recommend Nature Ecology & Evolution to your librarian <http://www.nature.com/subscriptions/recommend.html#forms>

** Visit the Springer Nature Editorial and Publishing website at www.springernature.com/editorial-and-publishing-jobs for more information about our career opportunities. If you have any questions please click here.**